# SENSITIVITY-AWARE DIFFERENTIALLY PRIVATE DECENTRALIZED LEARNING WITH ADAPTIVE NOISE

## ABSTRACT

Most existing decentralized learning methods with differential privacy (DP) employ fixed-level Gaussian noise during training, regardless of gradient convergence, which compromises model accuracy without providing additional privacy benefits. In this paper, we propose a novel Differentially Private Decentralized learning approach, termed AdaD$^2$P, which employs Adaptive noise leveraging the real-time estimation of sensitivity for local updates based on gradient norms and works for time-varying communication topologies. Compared with existing solutions, the integration of adaptive noise enables us to enhance model accuracy while preserving the $(\epsilon, \delta)$-DP privacy budget. We prove that AdaD$^2$P achieves a utility bound of $\mathcal{O}\left(\sqrt{d \log\left(\frac{1}{\delta}\right)}/(\sqrt{n}J\epsilon)\right)$, where $J$ and $n$ are the number of local samples and nodes, respectively, and $d$ the dimension of decision variable; this bound matches that of the distributed counterparts with server-client structures, without relying on the stringent bounded gradient assumption commonly used in previous works. Theoretical analysis reveals the inherent advantages of AdaD$^2$P employing adaptive noise as opposed to constant noise. Extensive experiments on two benchmark datasets demonstrate the superiority of AdaD$^2$P over its counterparts, especially under a strong level of privacy guarantee.

## 1 INTRODUCTION

Distributed learning has recently attracted significant attention due to its great potential in enhancing computing efficiency and has thus been widely adopted in various application domains (Langer et al., 2020). In particular, it can be typically modeled as a non-convex finite-sum optimization problem solved by a group of $n$ nodes, as depicted as follows:

$$\min_{x \in \mathbb{R}^d} f(x) \triangleq \frac{1}{n} \sum_{i=1}^{n} f_i(x), \quad \text{where } f_i(x) = \frac{1}{J} \sum_{j=1}^{J} f_i(x; j), \tag{1}$$

where $J$ denotes the local dataset size of each node, $f_i(x; j)$ denotes the loss function of the $j$-th data sample at node $i$ with respect to the model parameter $x \in \mathbb{R}^d$, and $f_i(x)$ and $f(x)$ denote the local objective function at node $i$ and the global objective function. All nodes collaborate to seek the optimal model parameter to minimize $f(x)$, and each node $i$ can only evaluate local stochastic gradient $\nabla f_i(x; \xi_i)$ where $\xi_i \in \{1, 2, ..., J\}$.

Bottlenecks such as high communication overhead and the vulnerability of central nodes in parameter server-based methods (Li et al., 2014; Zinkevich et al., 2010; McMahan et al., 2017a), motivate researchers to investigate fully decentralized methods (Lian et al., 2017; Tang et al., 2018; Lian et al., 2018) to solve Problem (1), where the central node is not required and each node only communicates with its neighbors. The existing decentralized learning algorithms usually employ undirected graphs for communication, which can not be easily implemented due to the existence of deadlocks (Assran et al., 2019). It is desirable to consider more practical scenarios where communication graphs may be directed and even time-varying. Stochastic gradient push (SGP) proposed in (Assran et al., 2019), which builds on push-sum protocol (Kempe et al., 2003), is proven to be very effective in solving (1) over directed and time-varying communication graphs.

In decentralized learning systems, all nodes frequently exchange information such as model parameters with their neighbors. This raises significant concerns about privacy, as the exposure of

intermediate parameters could potentially be exploited to compromise the privacy of original data samples (Wang et al., 2019b). To safeguard each node from potential data privacy attack, differential privacy (DP), as a theoretical tool to provide rigorous privacy guarantees and quantify privacy loss, can be integrated into each node within decentralized learning systems to enhance privacy protection.

Most existing decentralized learning algorithms with differential privacy guarantee for non-convex problems tend to either assume stochastic gradients are bounded by some constant $G$ (Yu et al., 2021; Xu et al., 2021) or employ gradient clipping strategy with a fixed clipping bound $C$ (Li & Chi, 2023), and they use constant $G$ or $C$ to estimate the $l_2$ sensitivity $S$ of gradient update across all iterations. As a result, each node injects fixed-level DP Gaussian noises with a variance proportional to the estimated sensitivity $S$ before performing local SGD at each iteration. However, our empirical observations indicate that the norm of gradient typically decay as training progresses and ultimately converges to a small value (c.f., Figure 1). This observation suggests that the aforementioned methods estimating $l_2$ sensitivity using constant $G$ or $C$ for all iterations may be conservative as gradient norms are often smaller than the constant $G$ or $C$, especially in the later stage of training. Therefore, their added fixed-level Gaussian noise deems unnecessary and will, instead, degrade the model accuracy without providing additional privacy gain. To this end, the following question arises naturally:

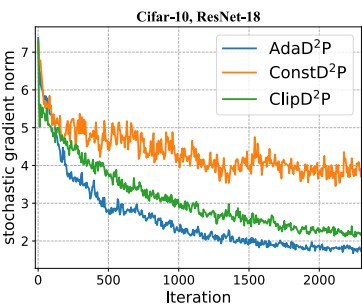

Figure 1: The evolution of gradient norm w.r.t the number of iterations for the proposed AdaD$^2$P and other two strategies (ConstD$^2$P and ClipD$^2$P) with constant noise.

> *"Can we design a decentralized learning method that adjusts the level of DP noise according to gradient norms during training while maintaining the privacy guarantee?"*

To address this question, we develop a new differentially private learning method for non-convex problems in fully decentralized settings, which can adapt the noise level to the actual privacy requirements as the training progresses and thus enhance model accuracy given the same privacy budget. The key contributions are summarized as follows:

- **New efficient algorithm with adaptive DP noise.** We propose a differentially private decentralized learning method with adaptive DP noise (termed AdaD$^2$P) for non-convex problems, which works for time-varying directed communication topologies. In particular, each node adds noise with a variance calculated according to the noise scale and the sensitivity estimated based on real-time gradient norms. This adaptive mechanism allows adding smaller noise and thus enhancing model accuracy without compromising privacy budgets; importantly, it can be readily integrated into other existing decentralized algorithms.

- **Theoretical analysis and utility guarantees.** We prove that AdaD$^2$P achieves a utility bound of $\mathcal{O}\left(\sqrt{d\log\left(\frac{1}{\delta}\right)}/(\sqrt{n}J\epsilon)\right)$, which matches that of existing distributed methods with server-client structures (c.f., Table 1). Our proof involves constructing an intricate loop among the utility gap captured by the running average squared gradient norm, consensus error and error terms arising from injected DP noise; importantly, the proof *does not rely on the restrictive bounded gradient assumption* as commonly used by the previous works. Besides, we provide theoretical evidence that sheds light on the inherent advantages of AdaD$^2$P employing adaptive noise compared to that with fixed-level noise.

- **Extensive experimental evaluations.** Extensive experiments conducted on training ResNet-18 DNN (resp. 2-layer neural network) on the Cifar-10 (resp. Mnist) dataset in fully decentralized setting show that, when adhering to a same privacy budget constraint, our proposed AdaD$^2$P achieves superior model accuracy compared to its counterparts that employ fixed-level Gaussian noise, particularly in the strong privacy protection region.

## 2   PRELIMINARY AND RELATED WORK

**Differential privacy.**   Differential privacy (DP) was originally introduced in the seminal work by Dwork et al.(Dwork et al., 2006) as a foundational concept for quantifying the privacy-preserving

| Algorithm | Privacy | Utility | Architecture | Without Assumption 6 |
|---|---|---|---|---|
| DP-SGD (Abadi et al., 2016) | $(\epsilon, \delta)$-DP | $\frac{\sqrt{d \log\left(\frac{1}{\delta}\right)}}{J\epsilon}$ | single node centralized | ✗ |
| Distributed DP-SRM[1] (Wang et al., 2019a) | $(\epsilon, \delta)$-DP global | $\frac{\sqrt{d \log\left(\frac{1}{\delta}\right)}}{nJ\epsilon}$ | $n$ nodes server-client | ✗ |
| LDP SVRG/SPIDER (Lowy et al., 2023) | $(\epsilon, \delta)$-DP for each node | $\frac{\sqrt{d \log\left(\frac{1}{\delta}\right)}}{\sqrt{n}J\epsilon}$ | $n$ nodes server-client | ✗ |
| SoteriaFL-SAGA/SVRG (Li et al., 2022) | $(\epsilon, \delta)$-DP for each node | $\frac{\sqrt{(1+\omega)d \log\left(\frac{1}{\delta}\right)}}{\sqrt{n}J\epsilon}$ | $n$ nodes server-client | ✗ |
| AdaD$^2$P (Algorithm 1) | $(\epsilon, \delta)$-DP for each node | $\frac{\sqrt{d \log\left(\frac{1}{\delta}\right)}}{\sqrt{n}J\epsilon}$ | $n$ nodes decentralized | ✔ |

[1] The global $(\epsilon, \delta)$-DP is considered therein, which only protects the privacy for the entire dataset while we consider $(\epsilon, \delta)$-DP for each node $i$, protecting the local dataset at the node's level.

Table 1: Comparison of existing differentially private stochastic algorithms for non-convex problems. Communication compression is employed in SoteriaFL-SAGA/SVRG with $\omega$ being the compression parameter. The Big $\mathcal{O}$ notation is omitted for simplicity.

capabilities of randomized algorithms. DP has now found widespread applications in a variety of domains that necessitate safeguarding against unintended information leakage, such as principle component analysis (Ge et al., 2018), meta learning (Li et al., 2019a), personalized recommendation (Shin et al., 2018), empirical risk minimization (Chaudhuri et al., 2011) and wireless network (Wei et al., 2021b). The standard definition of DP is provided as follows.

**Definition 1** (($\epsilon, \delta$)-DP (Dwork et al., 2014)). *A randomized mechanism $\mathcal{M}$ with domain $\mathcal{D}$ and range $\mathcal{R}$ satisfies $(\epsilon, \delta)$-differential privacy (or $(\epsilon, \delta)$-DP), if for any two adjacent inputs $\mathrm{x}, \mathrm{x}' \in \mathcal{D}$ differing on a single entry and for any subset of outputs $S \subseteq \mathcal{R}$, it holds that*

$$Pr\left[\mathcal{M}\left(\mathrm{x}\right) \in S\right] \leqslant e^\epsilon Pr\left[\mathcal{M}\left(\mathrm{x}'\right) \in S\right] + \delta, \tag{2}$$

*where the privacy budget $\epsilon$ denotes the privacy lower bound to measure a randomized query and $\delta$ is the probability of breaking this bound.*

A commonly employed technique to ensure a $(\epsilon, \delta)$-differential privacy guarantee is through the use of the Gaussian mechanism as provided below.

**Proposition 1** (Gaussian mechanism (Dwork et al., 2014)). *Let $f : \mathcal{D} \to \mathbb{R}$ be a real-valued function with $S$ being $f$'s $l_2$ sensitivity. Then, adding Gaussian noise $\mathcal{N}(0, \sigma^2 S^2)$ to $f$ such that $\mathcal{M}(\mathrm{x}) = f(\mathrm{x}) + \mathcal{N}(0, \sigma^2 S^2)$ satisfies $(\epsilon, \delta)$-DP if the noise scale $\sigma \geqslant \frac{\sqrt{2 \log(1.25/\delta)}}{\epsilon}$.*

The above proposition illustrates an inverse relationship between the noise scale $\sigma$ and privacy budget $\epsilon$ for a fixed $\delta$, and the fact that the noise variance $\sigma^2 S^2$ is dependent on both the noise scale $\sigma$ and $l_2$ sensitivity $S$. For iterative training processes, the cumulative privacy spending can be calculated using the basic composition theorem (Dwork et al., 2006; Dwork & Lei, 2009) and advanced composition theorem (Dwork et al., 2010; Bun & Steinke, 2016). To achieve a more precise estimate of the overall privacy budget throughout the entire training process, Abadi et al. (2016) introduced the moments accountant method that tracks higher moments. In the rest of this section, we will review existing research works related to achieving differential privacy guarantees in machine learning and highlight their limitations inherent in decentralized scenarios.

**Decentralized learning methods with privacy guarantee.** DP guarantee is initially integrated to centralized (single-node) setting for designing differentially private stochastic learning algorithms (Abadi et al., 2016; Wang et al., 2017; Iyengar et al., 2019; Chen et al., 2020; Wang et al., 2020). Further, DP guarantee is considered in distributed learning with server-client structures and the representative works include (McMahan et al., 2017b; Li et al., 2019b; Wang et al., 2019a; Wu

et al., 2020; Zhang et al., 2020; Wei et al., 2020; Zeng et al., 2021; Wei et al., 2021a; Ding et al., 2021; Li et al., 2022; Liu et al., 2022; Lowy et al., 2023; Wang et al., 2023; Zhou et al., 2023; Wei et al., 2023). Recently, there have been few works aiming to achieve DP guarantees for fully decentralized learning algorithms. For example, Cheng et al. (2018; 2019) achieve DP in fully decentralized learning for only strongly convex problems. Wang & Nedic (2022) achieve DP in fully decentralized architectures by tailoring gradient methods for deterministic optimization problems. For non-convex stochastic optimization problems as we consider in this work, Yu et al. (2021) present a differentially private decentralized learning method (DP$^2$-SGD) based on D-PSGD (Lian et al., 2017), which relies on a fixed communication topology and uses the basic composition theorem to bound the overall privacy loss. To have a tight privacy guarantee, Xu et al. (2021) propose a differentially private asynchronous decentralized learning method (A(DP)$^2$SGD) based on AD-PSGD (Lian et al., 2018), which provides privacy guarantee in the sense of Rényi differential privacy (RDP) (Mironov, 2017). However, it should be noted that the aforementioned two algorithms (Yu et al., 2021; Xu et al., 2021) work only for undirected communication graphs which is often not satisfied in practical scenarios, and they rely on the bounded gradient assumption. Most recently, Li & Chi (2023) achieve DP guarantee in decentralized learning for non-convex problems without bounded gradient assumption by employing gradient clipping strategy with a fixed clipping bound $C$, while their method is only applicable to time-invariant communication topologies.

**Learning with Adaptive DP Gaussian noise level.** For the aforementioned differentially private decentralized methods designed for non-convex stochastic optimization problems (Yu et al., 2021; Xu et al., 2021; Li & Chi, 2023), the injected noise level may exceeds what is actually needed for privacy requirements as training progresses, especially during the later stages of training, since their estimated sensitivity based on fixed $G$ (Yu et al., 2021; Xu et al., 2021) or $C$ (Li & Chi, 2023) may not reflect the actual value of sensitivity. The overestimate of sensitivity may, indeed, lead to a waste of unnecessary privacy budget during training process (Wei et al., 2023). There has been few works dedicated to precisely estimate the sensitivity in a real-time manner. For instance, a scheme of decaying gradient clipping bound has been employed to estimate the sensitivity in differentially private centralized learning (Du et al., 2021; Wei & Liu, 2021), yielding decreasing amount of noise injection. In the realm of distributed learning, the similar strategy of adaptive clipping bounds are utilized in (Andrew et al., 2021) to estimate the sensitivity. Most recently, Wei et al. (2023) use the minimum value of properly decaying clipping bound and current gradient norm to more accurately estimate the $l_2$ sensitivity, leading to a less amount of noise injection. However, these distributed methods (Andrew et al., 2021; Fu et al., 2022; Wei et al., 2023) only focus on the server-client architecture and no theoretical guarantee on model utility is provided therein. In contrast, we aim to design a differentially private decentralized learning method which incorporates adaptive noise levels in fully distributed settings and provide a rigorous theoretical utility guarantee.

## 3  PROPOSED ALGORITHM

We consider solving Problem (1) over the following general network model.

**Network Model.** The communication topology is modeled as a sequence of time-varying directed graph $\mathcal{G}^k = (\mathcal{V}, \mathcal{E}^k)$, where $\mathcal{V} = \{1, 2, ..., n\}$ denotes the set of nodes and $\mathcal{E}^k \subset \mathcal{V} \times \mathcal{V}$ denotes the set of directed edges/links at iteration $k$. We associate each graph $\mathcal{G}^k$ with a non-negative mixing matrix $P^k \in \mathbb{R}^{n \times n}$ such that $(i, j) \in \mathcal{E}^k$ if $P_{i,j}^k > 0$, i.e., node $i$ receiving a message from node $j$ at iteration $k$. Without loss of generality, we assume that each node is an in-neighbor of itself.

The following assumptions are made on the mixing matrix and graph for the above network model to facilitate the subsequent utility analysis for the proposed algorithm.

**Assumption 1** (Stochasicity of Mixing Matrix). *The non-negative mixing matrix $P^k, \forall k$ is column-stochastic, i.e., $\mathbf{1}^\top P^k = \mathbf{1}^\top$, where $\mathbf{1}$ is a vector with all of its elements equal to $1$.*

**Assumption 2** ($B$-strongly Connected). *There exists finite, positive integers $B$ and $\triangle$, such that the graph with edge set $\bigcup_{k=lB}^{(l+1)B-1} \mathcal{E}^k$ is strongly connected and has diameter at most $\triangle$ for $\forall l \geqslant 0$.*

**Algorithm Development.** Now we present our proposed AdaD$^2$P, a novel differentially private decentralized stochastic learning algorithm for non-convex problems with adaptive DP Gaussian noise level, which can work over general time-varying directed communication topologies; the complete pseudocode is summarized in Algorithm 1. At a high level, AdaD$^2$P is comprised of local SGD

and the averaging of neighboring information, following a framework similar to SGP (Assran et al., 2019). This framework involves the use of the Push-Sum protocol (Kempe et al., 2003), which can tackle the unblanceness of directed topologies by asymptotically estimating the Perron–Frobenius eigenvector of transition matrices. However, the key distinction lies in the injection of adaptive DP Gaussian noise before preforming local SGD. In particular, each node $i$ maintains three variables during the learning process: i) the model parameter $x_i^k$; ii) the scalar Push-Sum weight $w_i^k$ and iii) the de-biased parameter $z_i^k = x_i^k/w_i^k$, with the initialization of $x_i^0 = z_i^0 \in \mathbb{R}^d$ and $w_i^0 = 1$ for all nodes $i \in \mathcal{V}$. At each iteration $k$, each node $i$ updates as follows:

$$\underbrace{x_i^{k+\frac{1}{2}} = x_i^k - \gamma\left(\nabla f_i(z_i^k;\xi_i^k) + N_i^k\right),}_{\textbf{Differentially private local SGD}} \quad \underbrace{x_i^{k+1} = \sum_{j=1}^n P_{i,j}^k x_j^{k+\frac{1}{2}}, w_i^{k+1} = \sum_{j=1}^n P_{i,j}^k w_j^k,}_{\textbf{Neighboring information averaging}} \underbrace{z_i^{k+1} = \frac{x_i^{k+1}}{w_i^{k+1}},}_{\textbf{De-bias}}$$

where $\gamma > 0$ is the step size and $\nabla f_i(z_i^k;\xi_i^k)$ is the gradient evaluated on the de-biased parameter $z_i^k$ and training sample with index $\xi_i^k$ at node $i$. The injected randomized noise $N_i^k$ ensuring differential privacy guarantee for node $i$ is drawn from the Gaussian distribution (c.f., (3)) with a variance calculated according to the noise scale $\sigma$ and dynamic sensitivity estimated based on gradient norms It should be noted that gradient norm is a tighter estimation of actual sensitivity for noise injection than fixed $G$ and $C$ in most cases, especially at the later stage of training (Wei et al., 2023).

---

**Algorithm 1** Differentially Private Decentralized Learning with Adaptive Noise (**AdaD$^2$P**)

---

1: **Initialization:** $x_i^0 = z_i^0 \in \mathbb{R}^d$, $w_i^0 = 1$, step size $\gamma > 0$, total number of iterations $K$ and privacy budget $(\epsilon, \delta)$.
2: **for** $k = 0, 1, 2, ..., K-1$, at node $i$, **do**
3:     Randomly samples a local training data $\xi_i^k$ with the sampling probability $\frac{1}{J}$;
4:     Computes stochastic gradient at $z_i^k$: $\nabla f_i(z_i^k;\xi_i^k)$;
5:     Draws randomized noise $N_i^k$ from the Gaussian distribution

$$N_i^k \sim \mathcal{N}\left(0, \sigma^2\left\|\nabla f_i\left(z_i^k;\xi_i^k\right)\right\|^2 \mathbb{I}_d\right), \tag{3}$$

   where the noise scale $\sigma$ is defined in Proposition 2;
6:     Differentially private local SGD:

$$x_i^{k+\frac{1}{2}} = x_i^k - \gamma(\nabla f_i(z_i^k;\xi_i^k) + N_i^k); \tag{4}$$

7:     Sends $\left(x_i^{k+\frac{1}{2}}, w_i^k\right)$ to all out-neighbors ;
8:     Receives $\left(x_j^{k+\frac{1}{2}}, w_j^k\right)$ from all in-neighbors ;
9:     Updates $x_i^{k+1}$ by:   $x_i^{k+1} = \sum_{j=1}^n P_{i,j}^k x_j^{k+\frac{1}{2}}$ ;
10:     Updates $w_i^{k+1}$ by:   $w_i^{k+1} = \sum_{j=1}^n P_{i,j}^k w_j^k$ ;
11:     Updates $z_i^{k+1}$ by:   $z_i^{k+1} = x_i^{k+1}/w_i^{k+1}$ .
12: **end for**

---

**Remark 1.** *For comparison, we also present two counterparts ConstD$^2$P and ClipD$^2$P, which employ fixed-level noise with variance calculated according to fixed $l_2$ sensitivity, estimated using uniform gradient bound $G$ and fixed gradient clipping bound $C$ respectively. The complete pseudocodes of ConstD$^2$P and ClipD$^2$P can be found in Algorithm 2 and 3 in the appendix, respectively.*

## 4 THEORETICAL GUARANTEES

In this section, we provide the privacy and utility guarantee for our proposed AdaD$^2$P. In particular, we show that DP guarantee for each node can be achieved by setting the DP Gaussian noise scale $\sigma$ properly according to the given certain privacy budget $(\epsilon, \delta)$ and the total number of iterations $K$, which is summarized in the following proposition.

**Proposition 2** (Privacy guarantee). *There exist constants $c_1$ and $c_2$ such that, for any $\epsilon < \frac{c_1 K}{J^2}$ and $\delta \in (0, 1)$, $(\epsilon, \delta)$-DP can be guaranteed for each node $i$ for AdaD$^2$P, ConstD$^2$P and ClipD$^2$P after $K$ iterations if we set the noise scale*

$$\sigma = \frac{c_2 \sqrt{K \log\left(\frac{1}{\delta}\right)}}{J\epsilon}. \tag{5}$$

*Proof.* The proof of the above result can be easily adapted from Theorem 1 in (Abadi et al., 2016) by knowing the fact that the sampling probability is $\frac{1}{J}$ for each node $i$ at each iteration. $\square$

**Remark 2.** *The above theorem demonstrates that the variance of injected Gaussian noise for each node $i$ at each iteration $k$ for AdaD$^2$P is*

$$\mathbb{E}\left[\left\|N_i^k\right\|^2\right] \overset{(3)}{=} d\sigma^2 \left\|\nabla f_i\left(z_i^k; \xi_i^k\right)\right\|^2 \overset{(5)}{=} \underbrace{\frac{dc_2^2 \log\left(\frac{1}{\delta}\right)}{J^2\epsilon^2}}_{\beta} \cdot K \left\|\nabla f_i\left(z_i^k; \xi_i^k\right)\right\|^2, \tag{6}$$

*which is proportional to the real-time gradient norm $\left\|\nabla f_i\left(z_i^k; \xi_i^k\right)\right\|$.*

Next, we make the following blanket assumptions for the utility analysis of AdaD$^2$P.

**Assumption 3** (L-smooth). *For each function $f_i, i \in \mathcal{V}$, there exists a constant $L > 0$ such that $\left\|\nabla f_i\left(x\right) - \nabla f_i\left(y\right)\right\| \leqslant L \left\|x - y\right\|$.*

**Assumption 4** (Unbiased gradient). *For $\forall x \in \mathbb{R}^d$, the expectation of stochastic gradients of node $i$ is its aggregated gradient, i.e.,*

$$\mathbb{E}\left[\nabla f_i\left(x; \xi_i\right)\right] = \nabla f_i\left(x\right). \tag{7}$$

**Assumption 5** (Bounded variance). *There exist finite positive constants $\zeta^2$ and $b^2$ such that for any node $i$ and $\forall x \in \mathbb{R}^d$,*

$$\mathbb{E}\left[\left\|\nabla f_i\left(x; \xi_i\right) - \nabla f_i\left(x\right)\right\|\right] \leqslant \zeta^2 \tag{8}$$

*and*

$$\left\|\nabla f_i\left(x\right) - \nabla f\left(x\right)\right\|^2 \leqslant b^2. \tag{9}$$

With the above assumptions, by properly choosing the total number of iterations $K$ and the step size $\gamma$, we can obtain the utility guarantee of AdaD$^2$P (Algorithm 1) without relying on the bounded gradient assumption, which is presented in the following Theorem 1.

**Theorem 1** (Utility guarantee). *Suppose Assumptions 1-5 hold and $J \geqslant n^{\frac{3}{2}} c_2 \sqrt{d \log\left(\frac{1}{\delta}\right)}/\epsilon$ for a given privacy budget $(\epsilon, \delta)$. There exist constants $C$ and $q \in [0, 1)$, which depend on the diameter of the network $\triangle$ and the sequence of mixing matrices $P^k$, such that, if we set $\gamma = 1/(\frac{J\epsilon}{c_2 \sqrt{nd \log\left(\frac{1}{\delta}\right)}} + \hat{\gamma}(C, q)^{-1})$, $K = \frac{J^2 \epsilon^2}{dc_2^2 \log\left(\frac{1}{\delta}\right)}$ and the noise scale $\sigma = c_2 \sqrt{K \log\left(\frac{1}{\delta}\right)}/(J\epsilon)$, AdaD$^2$P can achieve $(\epsilon, \delta)$-DP guarantee for each node and has the following utility bound*

$$\frac{1}{K}\sum_{k=0}^{K-1} \mathbb{E}\left[\left\|\nabla f\left(\bar{x}^k\right)\right\|^2\right] \leqslant \mathcal{O}\left(\frac{\sqrt{d \log\left(\frac{1}{\delta}\right)}}{\sqrt{n} J\epsilon}\right), \tag{10}$$

*where $C$ and $q$ can be found in Lemma 4 and the definition of constant $\hat{\gamma}(C, q)$ can be found at (46) in the appendix, respectively. The Big $\mathcal{O}$ notation hides all constants involved in our setting, e.g., $L, \zeta, b, C, q, \sum_{i=1}^n \left\|x_i^0\right\|^2$ and $f\left(\bar{x}^0\right) - f^*$, where $f^* = \min_{x \in \mathbb{R}^d} f\left(x\right)$.*

*Proof.* The complete proof can be found in Section A.3 in the appendix. $\square$

**Remark 3.** *Table 1 provides a detailed comparison of our AdaD$^2$P with existing centralized/server-client algorithms, where the bounded gradient assumption is all assumed in their utility analysis, except our AdaD$^2$P. AdaD$^2$P achieves a utility bound of $\mathcal{O}\left(\sqrt{d \log\left(\frac{1}{\delta}\right)}/(\sqrt{n} J\epsilon)\right)$, matching that*

*of distributed methods with server-client structures, such as LDP SVRG/SPIDER, and SoteriaFL-SAGA/SVRG without communication compression ($\omega = 0$). Furthermore, AdaD$^2$P recovers the utility bound of centralized DP-SGD with $n = 1$. For completeness, we provide the derivation of the utility bound of the baseline centralized DP-SGD in Section A.6 in the appendix.*

Now, we provide the theoretical rationale behind the superior model performance of AdaD$^2$P in comparison to ConstD$^2$P under the same level of privacy protection. To this end, we split the upper bound of the metric (utility gap) into two distinct components: the standard term associated with SGP (Assran et al., 2019) and the term related to privacy noise, without specifying the value of $K$. As a result, we derive the following result for the proposed AdaD$^2$P algorithm.

**Proposition 3.** *Suppose Assumptions 1-5 hold. If the step size $\gamma \leqslant \min\left\{\frac{1-q}{6LC}, \frac{1}{L}\right\}$ and the noise scale $\sigma = c_2 \sqrt{K \log\left(\frac{1}{\delta}\right)} / (J\epsilon)$, AdaD$^2$P (Algorithm 1) can achieve $(\epsilon, \delta)$-DP guarantee for each node after $K$ iterations and has the following error bound*

$$\frac{1}{K}\sum_{k=0}^{K-1}\mathbb{E}\left[\left\|\nabla f\left(\bar{x}^k\right)\right\|^2\right] \leqslant \Xi + \underbrace{\left(\frac{12\gamma^2 L^2 C^2 \beta}{(1-q)^2} + \frac{2\gamma L\beta}{n}\right)\sum_{k=0}^{K-1}\frac{1}{n}\sum_{i=1}^{n}\mathbb{E}\left[\left\|\nabla f_i\left(z_i^k; \xi_i^k\right)\right\|^2\right]}_{\text{caused by adaptive privacy noise}},$$

(11)

*where $\Xi = \frac{4\left(f\left(\bar{x}^0\right) - f^*\right)}{\gamma K} + \frac{2\gamma L}{n}\zeta^2 + \frac{12\gamma^2 L^2 C^2}{(1-q)^2}\left(\zeta^2 + 3b^2\right) + \frac{12L^2 C^2}{(1-q)^2 nK}\sum_{i=1}^{n}\left\|x_i^0\right\|^2$ is the standard error term of SGP algorithm, and $\beta$ is defined at (6).*

*Proof.* The complete proof can be found in Section A.4 in the appendix. □

Next, we provide a similar result for ConstD$^2$P (Algorithm 2) which employs fixed-level noise relying on the following bounded gradient assumption.

**Assumption 6** (Bounded gradient). *For any $z \in \mathbb{R}^d$ and $\xi_i \in \{1, 2, ..., J\}$, there exists finite positive constant $G$ such that*

$$\left\|\nabla f_i\left(z; \xi_i\right)\right\| \leqslant G. \tag{12}$$

**Proposition 4.** *Under the same condition of Proposition 3 and suppose Assumption 6 holds, ConstD$^2$P (Algorithm 2) can achieve $(\epsilon, \delta)$-DP guarantee for each node after $K$ iterations and*

$$\frac{1}{K}\sum_{k=0}^{K-1}\mathbb{E}\left[\left\|\nabla f\left(\bar{x}^k\right)\right\|^2\right] \leqslant \Xi + \underbrace{\left(\frac{12\gamma^2 L^2 C^2 \beta}{(1-q)^2} + \frac{2\gamma L\beta}{n}\right)\sum_{k=0}^{K-1}\frac{1}{n}\sum_{i=1}^{n}G^2}_{\text{caused by fixed privacy noise}}. \tag{13}$$

*Proof.* The complete proof can be found in Section A.5 in the appendix. □

**Remark 4** (Insights for the adavantage of AdaD$^2$P). *The comparison between the upper bounds in (11) and (13) reveals a significant difference in the components related to privacy noise. In particular, as the stochastic gradient norm tends to decay throughout the training process, it becomes evident that the component related to privacy noise in AdaD$^2$P is much tighter compared to that of ConstD$^2$P. This insight sheds light on the reason behind AdaD$^2$P outperforming ConstD$^2$P under the same level of privacy protection, as demonstrated in the experiments in Section 5.*

## 5 EXPERIMENTS

We conduct several experiments to verify the performance of AdaD$^2$P (Algorithm 1), with comparison to the counterparts algorithms ConstD$^2$P (Algorithm 2) and ClipD$^2$P (Algorithm 3) which both employ fixed-level noise. All experiments are deployed in a high performance server with Intel Xeon E5-2680 v4 CPU @ 2.40GHz and 8 Nvidia RTX 3090 GPUs, and are implemented with distributed communication package *torch.distributed* in PyTorch (Paszke et al., 2017), where a process serves as a node, and inter-process communication is used to mimic communication among nodes.

**Experimental setup.** We compare three algorithms in fully decentralized setting composed of 20 nodes, on two non-convex learning tasks (i.e., deep CNN ResNet-18 training and shallow 2-layer

neural network training). For all experiments, we split shuffled datasets evenly to 20 nodes and use time-varying directed exponential graph (refer to Section C in the appendix for its definition) as communication topology for three algorithms. The learning rate is set as 0.05 for ResNet-18 training and 0.03 for 2-layer neural network training. It is worth noting that bounded gradient (Assumption 6) is required for ConstD$^2$P (Algorithm 2). To obtain this upper bound $G$, we run non-private SGP algorithm (no privacy noise) 5 times in advance and use maximum norm of stochastic gradient of the training process to be the estimate of $G$. In addition, according to Proposition 2, we know that when fixing $\delta$ (usually set as $10^{-5}$), the privacy budget $\epsilon$ depends on the noise scale $\sigma$ and the total number of iterations $K$. That is to say, if we run three algorithms for the same total iterations with the same noise scale $\sigma$, the privacy protection levels for three algorithms are the same.

## 5.1 DEEP CNN RESNET-18 TRAINING

The first task is to train CNN model ResNet-18 (He et al., 2016) on Cifar-10 dataset (Krizhevsky et al., 2009). In this setting, the value of $G$ is estimated to be $8.5$ using our aforementioned approach. For ClipD$^2$P (Algorithm 3), we test the fixed clipping bound $C$ with three different values chosen from the set $\{2, 3, 5\}$. We run three algorithms for 3500 iterations.

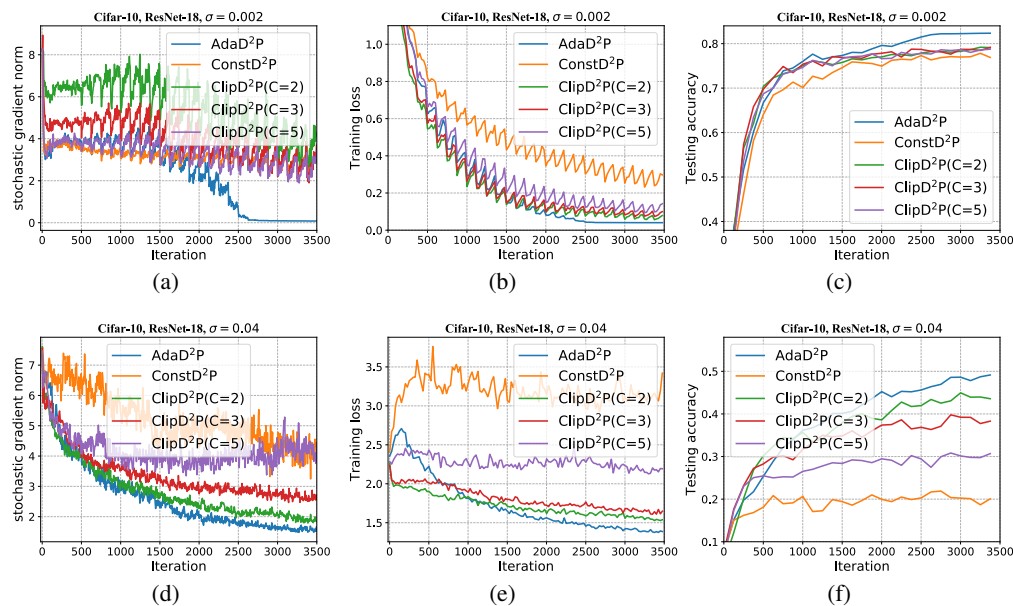

Figure 2: Performance comparison of training ResNet-18 for AdaD$^2$P with ConstD$^2$P and ClipD$^2$P under the same noise scale: $\sigma = 0.002$ for (a) (b) (c); $\sigma = 0.04$ for (d) (e) (f).

**Performance comparison under the same level of privacy protection.** We first set a relatively small $\sigma = 0.002$ for all three algorithms, which indicates a relatively modest level of privacy protection. The results depicted in Figures 2(a), 2(b) and 2(c) illustrate that, AdaD$^2$P outperforms the other two algorithms, in terms of the convergence of gradient norm, training loss and model accuracy. It is evident that when approaching the end of training, the gradient norm converges to a very small value near 0 for AdaD$^2$P, which results in a very minor amount of added noise, contributing positively to model accuracy further. In contrast, the other two algorithms inject fixed-level noise even during the later stages of the training process, leading to a degradation in model accuracy. -When setting a relatively larger $\sigma = 0.04$ which implies a relatively higher level of privacy protection, it follows from Figures 2(d), 2(e) and 2(f) that AdaD$^2$P still outperforms the other two algorithms, and shows more pronounced advantage in model accuracy (achieves a $30\%$ higher model accuracy than ConstD$^2$P). In appendix D, we present additional experimental results of using other values of $\sigma$, and we have the same experimental observations.

**Trade off between model utility and privacy protection level.** We vary the value of noise scale $\sigma$ from the set $\{0.001, 0.002, 0.005, 0.01, 0.02, 0.03, 0.04\}$, for AdaD$^2$P. The results presented in

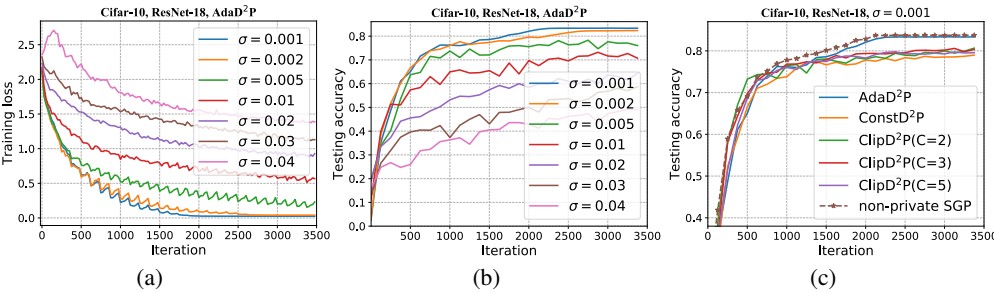

(a)  (b)  (c)

Figure 3: Performance comparison of training ResNet-18 for AdaD$^2$P under different noise scale $\sigma$ in terms of (a) training loss and (b) testing accuracy; (c) Performance comparison of three algorithms with non-private (no noise) SGP algorithm under the noise scale $\sigma = 0.001$
.

Figures 3(a) and 3(b) show that, as noise scale $\sigma$ increases which imply stronger privacy protection, the model utility (testing accuracy) deteriorates, illustrating the trade off between model utility and privacy protection level. Moreover, it follows from Figure 3(c) that when the noise scale is set as $\sigma = 0.001$, AdaD$^2$P is able to achieve model performance *almost without accuracy loss* compared to non-private SGP, while ConstD$^2$P and ClipD$^2$P still suffer from significant accuracy loss.

## 5.2 SHALLOW 2-LAYER NEURAL NETWORK TRAINING

Next we consider a simple shallow 2-layer neural network training task on Mnist (Deng, 2012) dataset. For this task, the value of $G$ is estimated to be 3.5 using the same approach. We set the clipping bound $C = 1$ for ClipD$^2$P and run three algorithms for the same 2200 iterations, and compare their performance under the same noise scale $\sigma$. It follows from the experimental results in Figure 4 that: under the same level of privacy protection (same $\sigma$), AdaD$^2$P outperforms the other two algorithms, and the advantage in model accuracy becomes more pronounced at a higher level of privacy protection (larger $\sigma$), verifying the superior performance of adaptive noise mechanism. Additional experimental tests for various $\sigma$ values are provided in appendix D, and we can observe the same experimental phenomenon.

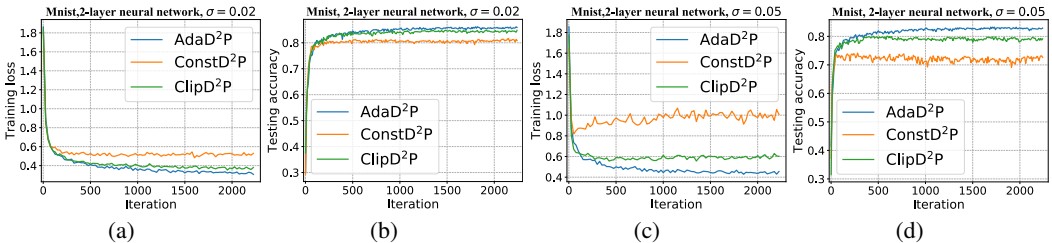

(a)  (b)  (c)  (d)

Figure 4: Performance comparison of training 2-layer neural network for AdaD$^2$P with ConstD$^2$P and ClipD$^2$P under the same noise scale: $\sigma = 0.02$ for (a) (b); $\sigma = 0.05$ for (c) (d).

## 6 CONCLUSION

In this paper, we proposed a differentially private decentralized learning method for non-convex problems (termed AdaD$^2$P), which employs adaptive noise level and works for general time-varying communication topologies. Without relying on the bounded gradient assumption, we proved that AdaD$^2$P achieves a utility bound which matches that of distributed counterparts with server-client structures. Theoretical analysis revealed the inherent advantages of AdaD$^2$P employing adaptive noise as opposed to constant noise. We conducted extensive experiments to verify the superior performance of AdaD$^2$P compared to its counterparts which employ fixed-level noise.

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
