# Appendix

## CONTENTS

## A  PROOF OF MAIN RESULTS

To facilitate our analysis, we first rewrite the $9^{th}$ step of the proposed AdaD$^2$P (c.f., Algorithm 1) in a compact form:

$$X^{k+1} = \left( X^k - \gamma \left( \nabla F \left( Z^k; \xi^k \right) + N^k \right) \right) \left( P^k \right)^\top \tag{14}$$

where $\left( P^k \right)^\top \in \mathbb{R}^{n \times n}$ is the transpose of the mixing matrix $P^k$ at iteration $k$, and

$X^k := \left[ x_1^k, x_2^k, \cdots, x_n^k \right] \in \mathbb{R}^{d \times n}$: concatenation of all the nodes' parameters at iteration $k$;

$Z^k := \left[ z_1^k, z_2^k, \cdots, z_n^k \right] \in \mathbb{R}^{d \times n}$: concatenation of all the nodes' de-biased parameters at iteration $k$;

$\nabla F(Z^k; \xi^k) := \left[ \nabla f_1(z_1^k; \xi_1^k), \nabla f_2(z_2^k; \xi_2^k), ..., \nabla f_n(z_n^k; \xi_n^k) \right] \in \mathbb{R}^{d \times n}$: concatenation of all the nodes' stochastic gradients at iteration $k$;

$N^k := \left[ N_1^k, N_2^k, ..., N_n^k \right] \in \mathbb{R}^{d \times n}$: concatenation of all the nodes' added Gaussian noise at iteration $k$.

Now, let $\bar{x}^k = \frac{1}{n} X^k \mathbf{1} = \frac{1}{n} \sum_{i=1}^n x_i^k \in \mathbb{R}^d$ denote the average of all nodes' parameters at iteration $k$. Then, the update of average system of (14) becomes

$$\bar{x}^{k+1} = \bar{x}^k - \gamma \cdot \left( \frac{1}{n} \sum_{i=1}^n \nabla f_i(z_i^k; \xi_i^k) + \frac{1}{n} \sum_{i=1}^n N_i^k \right) \tag{15}$$

which can be easily obtained by right multiplying $\frac{1}{n} \mathbf{1}$ from both sides of (14) and using the column-stochastic property of $P^k$ (c.f., Assumption 1). The above average system will be useful in subsequent analysis.

In addition, we denote $\mathcal{F}^k := \left\{ \bigcup_{i=1}^n \left( x_i^0, z_i^0, \xi_i^0, N_i^0, \cdots, x_i^{k-1}, z_i^{k-1}, \xi_i^{k-1}, N_i^{k-1}, x_i^k, z_i^k \right) \right\}$ as filtration of the history sequence upto $k$, and define $\mathbb{E}\left[ \cdot \left| \mathcal{F}^k \right. \right]$ the conditional expectation given $\mathcal{F}^k$.

**Outline of Proof.** Our proof of utility guarantee for AdaD$^2$P (c.f., Theorem 1) consists of three parts: i) we first provide several technical lemmas to facilitate the subsequent analysis (c.f., Section A.1); ii) we then provide two supporting lemmas to establish two key inequalities (c.f., Section A.2) where the first inequality is obtained by applying descent lemma (c.f., (18)) and the second inequality is to upper bound the consensus error (c.f.,(26)); iii) we finally upper bound the error term arising from injected DP noise to obtain the third key inequality (c.f., (34)). As a result, the utility bound can be derived by constructing a loop for the above three key inequalities with proper choice of certain parameters such as the step size $\gamma$ and the total number of iterations $K$ (c.f., Section A.3).

### A.1 IMPORTANT UPPER BOUNDS

In this section, we first provide several technical lemmas to facilitate the subsequent analysis.

**Lemma 1.** *Let $\left\{ v^k \right\}_{k=0}^\infty$ be a non-negative sequence and $\lambda \in (0,1)$. Then, we have*

$$\left( \sum_{l=0}^k \lambda^{k-l} v^l \right)^2 \leqslant \frac{1}{1-\lambda} \sum_{l=0}^k \lambda^{k-l} \left( v^l \right)^2. \tag{16}$$

*Proof.* Using Cauchy-Swarchz inequality, we have

$$\left( \sum_{l=0}^k \lambda^{k-l} v^l \right)^2 = \left( \sum_{l=0}^k \lambda^{\frac{k-l}{2}} \left( \lambda^{\frac{k-l}{2}} v^l \right) \right)^2$$

$$\leqslant \sum_{l=0}^k \left( \lambda^{\frac{k-l}{2}} \right)^2 \cdot \sum_{l=0}^k \left( \lambda^{\frac{k-l}{2}} v^l \right)^2$$

$$\leqslant \frac{1}{1-\lambda} \sum_{l=0}^k \lambda^{k-l} \left( v^l \right)^2,$$

which completes the proof. $\square$

**Lemma 2.** *Suppose Assumptions 3 and 5 hold. Then, we have*

$$\left\| \nabla f_i \left( z_i^k \right) \right\|^2 \leqslant 3L^2 \left\| z_i^k - \bar{x}^k \right\|^2 + 3b^2 + 3 \left\| \nabla f \left( \bar{x}^k \right) \right\|^2. \tag{17}$$

*Proof.* Using Assumption 3 ($L$-smooth) and Assumption 5, we have

$$\left\| \nabla f_i \left( z_i^k \right) \right\|^2 = \left\| \nabla f_i \left( z_i^k \right) - \nabla f_i \left( \bar{x}^k \right) + \nabla f_i \left( \bar{x}^k \right) - \nabla f \left( \bar{x}^k \right) + \nabla f \left( \bar{x}^k \right) \right\|^2$$

$$\leqslant 3 \left\| \nabla f_i \left( z_i^k \right) - \nabla f_i \left( \bar{x}^k \right) \right\|^2 + 3 \left\| \nabla f_i \left( \bar{x}^k \right) - \nabla f \left( \bar{x}^k \right) \right\|^2 + 3 \left\| \nabla f \left( \bar{x}^k \right) \right\|^2$$

$$\leqslant 3L^2 \left\| z_i^k - \bar{x}^k \right\|^2 + 3b^2 + 3 \left\| \nabla f \left( \bar{x}^k \right) \right\|^2,$$

which completes the proof. $\square$

## A.2 SUPPORTING LEMMAS

The following lemma is crucial to the convergence analysis in the non-convex stochastic optimization, which is obtained by applying the descent lemma recursively from $k = 0$ to $K$.

**Lemma 3.** *Suppose Assumption 3, 4 and 5 hold. For a given constant step size $\gamma$, we have*

$$
\frac{\gamma}{2} \sum_{k=0}^{K-1} \mathbb{E}\left[\left\|\nabla f\left(\bar{x}^k\right)\right\|^2\right] + \frac{\gamma\left(1 - \gamma L\right)}{2} \sum_{k=0}^{K-1} \mathbb{E}\left[\left\|\frac{1}{n}\sum_{i=1}^{n} \nabla f_i\left(z_i^k\right)\right\|^2\right]
$$

$$
\leqslant f\left(\bar{x}^0\right) - f^* + \frac{\gamma L^2}{2} \sum_{k=0}^{K-1} \frac{1}{n} \sum_{i=1}^{n} \mathbb{E}\left[\left\|z_i^k - \bar{x}^k\right\|^2\right] + \frac{\gamma^2 LK}{2n}\zeta^2 \tag{18}
$$

$$
+ \frac{\gamma^2 L}{2n} \cdot \underbrace{\beta K \sum_{k=0}^{K-1} \frac{1}{n} \sum_{i=1}^{n} \mathbb{E}\left[\left\|\nabla f_i(z_i^k; \xi_i^k)\right\|^2\right]}_{\text{caused by injected DP noise}}.
$$

*Proof.* Applying the descent lemma to $f$ at $\bar{x}^k$ and $\bar{x}^{k+1}$, we have

$$
f\left(\bar{x}^{k+1}\right) \leqslant f\left(\bar{x}^k\right) + \left\langle \nabla f\left(\bar{x}^k\right), \bar{x}^{k+1} - \bar{x}^k \right\rangle + \frac{L}{2}\left\|\bar{x}^{k+1} - \bar{x}^k\right\|^2
$$

$$
\overset{(15)}{=} f\left(\bar{x}^k\right) - \gamma \left\langle \nabla f\left(\bar{x}^k\right), \frac{1}{n}\sum_{i=1}^{n} \nabla f_i(z_i^k; \xi_i^k) + \frac{1}{n}\sum_{i=1}^{n} N_i^k \right\rangle
$$

$$
+ \frac{\gamma^2 L}{2}\left\|\frac{1}{n}\sum_{i=1}^{n} \nabla f_i(z_i^k; \xi_i^k) + \frac{1}{n}\sum_{i=1}^{n} N_i^k\right\|^2
$$

$$
= f\left(\bar{x}^k\right) - \gamma \left\langle \nabla f\left(\bar{x}^k\right), \frac{1}{n}\sum_{i=1}^{n} \nabla f_i(z_i^k; \xi_i^k) + \frac{1}{n}\sum_{i=1}^{n} N_i^k \right\rangle + \frac{\gamma^2 L}{2}\left\|\frac{1}{n}\sum_{i=1}^{n} \nabla f_i(z_i^k; \xi_i^k)\right\|^2
$$

$$
+ \frac{\gamma^2 L}{2}\left\|\frac{1}{n}\sum_{i=1}^{n} N_i^k\right\|^2 + \gamma^2 L \left\langle \frac{1}{n}\sum_{i=1}^{n} \nabla f_i(z_i^k; \xi_i^k), \frac{1}{n}\sum_{i=1}^{n} N_i^k \right\rangle.
$$

Taking the expectation of both sides conditioned on $\mathcal{F}^k$ for the above inequality, we obtain

$$
\mathbb{E}\left[f\left(\bar{x}^{k+1}\right)\big|\mathcal{F}^k\right]
$$

$$
\leqslant f\left(\bar{x}^k\right) - \gamma \mathbb{E}\left[\mathbb{E}\left[\left\langle \nabla f\left(\bar{x}^k\right), \frac{1}{n}\sum_{i=1}^{n} \nabla f_i(z_i^k; \xi_i^k) + \frac{1}{n}\sum_{i=1}^{n} N_i^k \right\rangle \big|\mathcal{F}^k, \xi^k\right]\big|\mathcal{F}^k\right]
$$

$$
+ \frac{\gamma^2 L}{2}\mathbb{E}\left[\left\|\frac{1}{n}\sum_{i=1}^{n} \nabla f_i(z_i^k; \xi_i^k)\right\|^2 \big|\mathcal{F}^k\right] + \frac{\gamma^2 L}{2}\mathbb{E}\left[\mathbb{E}\left[\left\|\frac{1}{n}\sum_{i=1}^{n} N_i^k\right\|^2 \big|\mathcal{F}^k, \xi^k\right]\big|\mathcal{F}^k\right]
$$

$$
+ \gamma^2 L \mathbb{E}\left[\mathbb{E}\left[\left\langle \frac{1}{n}\sum_{i=1}^{n} \nabla f_i(z_i^k; \xi_i^k), \frac{1}{n}\sum_{i=1}^{n} N_i^k \right\rangle \big|\mathcal{F}^k, \xi^k\right]\big|\mathcal{F}^k\right] \tag{19}
$$

$$
\overset{(7)}{=} f\left(\bar{x}^k\right) \underbrace{- \gamma \left\langle \nabla f\left(\bar{x}^k\right), \frac{1}{n}\sum_{i=1}^{n} \nabla f_i\left(z_i^k\right) \right\rangle}_{A_1} + \underbrace{\frac{\gamma^2 L}{2}\mathbb{E}\left[\left\|\frac{1}{n}\sum_{i=1}^{n} \nabla f_i(z_i^k; \xi_i^k)\right\|^2 \big|\mathcal{F}^k\right]}_{A_2}
$$

$$
+ \underbrace{\frac{\gamma^2 L}{2}\mathbb{E}\left[\mathbb{E}\left[\left\|\frac{1}{n}\sum_{i=1}^{n} N_i^k\right\|^2 \big|\mathcal{F}^k, \xi^k\right]\big|\mathcal{F}^k\right]}_{A_3}.
$$

For $A_1$ in (19), we have

$$
\begin{aligned}
A_1 &= -\frac{\gamma}{2}\left\|\nabla f\left(\bar{x}^k\right)\right\|^2 - \frac{\gamma}{2}\left\|\frac{1}{n}\sum_{i=1}^n \nabla f_i\left(z_i^k\right)\right\|^2 + \frac{\gamma}{2}\left\|\frac{1}{n}\sum_{i=1}^n \nabla f_i\left(z_i^k\right) - \nabla f\left(\bar{x}^k\right)\right\|^2 \\
&= -\frac{\gamma}{2}\left\|\nabla f\left(\bar{x}^k\right)\right\|^2 - \frac{\gamma}{2}\left\|\frac{1}{n}\sum_{i=1}^n \nabla f_i\left(z_i^k\right)\right\|^2 + \frac{\gamma}{2}\left\|\frac{1}{n}\sum_{i=1}^n \left(\nabla f_i\left(z_i^k\right) - \nabla f_i\left(\bar{x}^k\right)\right)\right\|^2 \\
&\leqslant -\frac{\gamma}{2}\left\|\nabla f\left(\bar{x}^k\right)\right\|^2 - \frac{\gamma}{2}\left\|\frac{1}{n}\sum_{i=1}^n \nabla f_i\left(z_i^k\right)\right\|^2 + \frac{\gamma}{2n}\sum_{i=1}^n \left\|\nabla f_i\left(z_i^k\right) - \nabla f_i\left(\bar{x}^k\right)\right\|^2 \\
&\stackrel{(a)}{\leqslant} -\frac{\gamma}{2}\left\|\nabla f\left(\bar{x}^k\right)\right\|^2 - \frac{\gamma}{2}\left\|\frac{1}{n}\sum_{i=1}^n \nabla f_i\left(z_i^k\right)\right\|^2 + \frac{\gamma L^2}{2n}\sum_{i=1}^n \left\|z_i^k - \bar{x}^k\right\|^2,
\end{aligned}
\tag{20}
$$

where in (a) we used Assumption 3.

For $A_2$ in (19), we have

$$
\begin{aligned}
A_2 &= \mathbb{E}\left[\left\|\frac{1}{n}\sum_{i=1}^n \left(\nabla f_i(z_i^k;\xi_i^k) - \nabla f_i(z_i^k)\right) + \frac{1}{n}\sum_{i=1}^n \nabla f_i(z_i^k)\right\|^2 \Big| \mathcal{F}^k\right] \\
&\stackrel{(7)}{=} \mathbb{E}\left[\left\|\frac{1}{n}\sum_{i=1}^n \nabla f_i(z_i^k)\right\|^2 \Big| \mathcal{F}^k\right] + \frac{1}{n^2}\mathbb{E}\left[\left\|\sum_{i=1}^n \left(\nabla f_i(z_i^k;\xi_i^k) - \nabla f_i(z_i^k)\right)\right\|^2 \Big| \mathcal{F}^k\right] \\
&= \left\|\frac{1}{n}\sum_{i=1}^n \nabla f_i(z_i^k)\right\|^2 + \frac{1}{n^2}\sum_{i=1}^n \mathbb{E}\left[\left\|\nabla f_i(z_i^k;\xi_i^k) - \nabla f_i\left(z_i^k\right)\right\|^2 \Big| \mathcal{F}^k\right] \\
&\stackrel{(8)}{\leqslant} \left\|\frac{1}{n}\sum_{i=1}^n \nabla f_i(z_i^k)\right\|^2 + \frac{\zeta^2}{n}.
\end{aligned}
\tag{21}
$$

For $A_3$ in (19), we have

$$
\begin{aligned}
A_3 &\stackrel{(6)}{=} \mathbb{E}\left[\frac{1}{n^2}\sum_{i=1}^n \beta K \left\|\nabla f_i(z_i^k;\xi_i^k)\right\|^2 \Big| \mathcal{F}^k\right] \\
&= \frac{\beta K}{n^2}\sum_{i=1}^n \mathbb{E}\left[\left\|\nabla f_i(z_i^k;\xi_i^k)\right\|^2 \Big| \mathcal{F}^k\right].
\end{aligned}
\tag{22}
$$

Then, substituting (20), (21) and (22) into (19) yields

$$
\begin{aligned}
&\mathbb{E}\left[f\left(\bar{x}^{k+1}\right) \big| \mathcal{F}^k\right] \\
&\leqslant f\left(\bar{x}^k\right) - \frac{\gamma}{2}\left\|\nabla f\left(\bar{x}^k\right)\right\|^2 - \frac{\gamma\left(1-\gamma L\right)}{2}\left\|\frac{1}{n}\sum_{i=1}^n \nabla f_i\left(z_i^k\right)\right\|^2 + \frac{\gamma L^2}{2n}\sum_{i=1}^n \left\|z_i^k - \bar{x}^k\right\|^2 \\
&\quad + \frac{\gamma^2 L \zeta^2}{2n} + \frac{\gamma^2 L}{2}\cdot\frac{\beta K}{n^2}\sum_{i=1}^n \mathbb{E}\left[\left\|\nabla f_i(z_i^k;\xi_i^k)\right\|^2 \big| \mathcal{F}^k\right].
\end{aligned}
\tag{23}
$$

Taking total expectation on both sides of (23), yields

$$
\begin{aligned}
\mathbb{E}\left[f\left(\bar{x}^{k+1}\right)\right] &\leqslant \mathbb{E}\left[f\left(\bar{x}^k\right)\right] - \frac{\gamma}{2}\mathbb{E}\left[\left\|\nabla f\left(\bar{x}^k\right)\right\|^2\right] - \frac{\gamma\left(1-\gamma L\right)}{2}\mathbb{E}\left[\left\|\frac{1}{n}\sum_{i=1}^n \nabla f_i\left(z_i^k\right)\right\|^2\right] \\
&\quad + \frac{\gamma L^2}{2n}\sum_{i=1}^n \mathbb{E}\left[\left\|z_i^k - \bar{x}^k\right\|^2\right] + \frac{\gamma^2 L \zeta^2}{2n} + \frac{\gamma^2 L}{2}\cdot\frac{\beta K}{n^2}\sum_{i=1}^n \mathbb{E}\left[\left\|\nabla f_i(z_i^k;\xi_i^k)\right\|^2\right].
\end{aligned}
$$

Summing the above inequality form $k=0$ to $K-1$, we obtain (18), which completes the proof. $\quad\square$

The following lemma bounds the distance between the de-biased parameters $z_i^k$ at each node $i$ and the node-wise average $\bar{x}^k$, which can be adapted from Lemma 3 in (Assran et al., 2019).

**Lemma 4.** *Suppose that Assumptions 1 and 2 hold. Let $\varepsilon$ be the minimum of all non-zero mixing weights, $\lambda = 1 - n\varepsilon^{\triangle B}$ and $q = \lambda^{\frac{1}{\triangle B + 1}}$. Then, there exists a constant*

$$C < \frac{2\sqrt{d}\varepsilon^{-\triangle B}}{\lambda^{\frac{\triangle B + 2}{\triangle B + 1}}}, \tag{24}$$

*such that for any $i \in \mathcal{V}$ and $k \geqslant 0$, we have*

$$\left\| z_i^k - \bar{x}^k \right\| \leqslant Cq^k \left\| x_i^0 \right\| + \gamma C \sum_{s=0}^{k} q^{k-s} \left\| \nabla f_i \left( z_i^s; \xi_i^s \right) + N_i^s \right\|. \tag{25}$$

Now, we attempt to upper bound the accumulative consensus error $\sum_{k=0}^{K-1} \frac{1}{n} \sum_{i=1}^{n} \mathbb{E} \left[ \left\| z_i^k - \bar{x}^k \right\|^2 \right]$ using $\sum_{k=0}^{K-1} \mathbb{E} \left[ \left\| \nabla f \left( \bar{x}^k \right) \right\|^2 \right]$, which is summarized in the following lemma.

**Lemma 5.** *Suppose Assumptions 1-5 hold. Then, we have*

$$\left( 1 - \frac{9\gamma^2 L^2 C^2}{(1-q)^2} \right) \sum_{k=0}^{K-1} \cdot \frac{1}{n} \sum_{i=1}^{n} \mathbb{E} \left[ \left\| z_i^k - \bar{x}^k \right\|^2 \right]$$

$$\leqslant \frac{9\gamma^2 C^2}{(1-q)^2} \sum_{k=0}^{K-1} \mathbb{E} \left[ \left\| \nabla f \left( \bar{x}^k \right) \right\|^2 \right] + K \cdot \frac{3\gamma^2 C^2}{(1-q)^2} \left( \zeta^2 + 3b^2 \right)$$

$$+ \frac{3C^2}{(1-q)^2 n} \sum_{i=1}^{n} \left\| x_i^0 \right\|^2 + \underbrace{\frac{3\gamma^2 C^2}{(1-q)^2} \cdot \beta K \sum_{k=0}^{K-1} \frac{1}{n} \sum_{i=1}^{n} \mathbb{E} \left[ \left\| \nabla f_i (z_i^k; \xi_i^k) \right\|^2 \right]}_{\text{caused by injected DP noise}}. \tag{26}$$

*Proof.* According to (25), we have

$$\left\| z_i^k - \bar{x}^k \right\| \leqslant Cq^k \left\| x_i^0 \right\| + \gamma C \sum_{s=0}^{k} q^{k-s} \left\| \nabla f_i \left( z_i^s; \xi_i^s \right) \right\| + \gamma C \sum_{s=0}^{k} q^{k-s} \left\| N_i^s \right\|. \tag{27}$$

Squaring on both sides of (27), we have

$$\left\| z_i^k - \bar{x}^k \right\|^2 \leqslant \left( Cq^k \left\| x_i^0 \right\| + \gamma C \sum_{s=0}^{k} q^{k-s} \left\| \nabla f_i \left( z_i^s; \xi_i^s \right) \right\| + \gamma C \sum_{s=0}^{k} q^{k-s} \left\| N_i^s \right\| \right)^2$$

$$\leqslant 3C^2 q^{2k} \left\| x_i^0 \right\|^2 + 3\gamma^2 C^2 \left( \sum_{s=0}^{k} q^{k-s} \left\| \nabla f_i \left( z_i^s; \xi_i^s \right) \right\| \right)^2 + 3\gamma^2 C^2 \left( \sum_{s=0}^{k} q^{k-s} \left\| N_i^s \right\| \right)^2$$

$$\leqslant 3C^2 q^{2k} \left\| x_i^0 \right\|^2 + \frac{3\gamma^2 C^2}{1-q} \sum_{s=0}^{k} q^{k-s} \left\| \nabla f_i \left( z_i^s; \xi_i^s \right) \right\|^2 + \frac{3\gamma^2 C^2}{1-q} \sum_{s=0}^{k} q^{k-s} \left\| N_i^s \right\|^2, \tag{28}$$

where we used $(a + b + c)^2 \leqslant 3a^2 + 3b^2 + 3c^2$ in the second inequality and Lemma 1 in the last inequality, respectively.

Taking total expectation on both sides of (28) yields

$$
\begin{aligned}
&\mathbb{E}\left[\left\|z_i^k - \bar{x}^k\right\|^2\right] \\
&\leqslant 3C^2 q^{2k}\left\|x_i^0\right\|^2 + \frac{3\gamma^2 C^2}{1-q}\sum_{s=0}^{k} q^{k-s}\mathbb{E}\left[\left\|\nabla f_i\left(z_i^s;\xi_i^s\right)\right\|^2\right] + \frac{3\gamma^2 C^2}{1-q}\sum_{s=0}^{k} q^{k-s}\mathbb{E}\left[\left\|N_i^s\right\|^2\right] \\
&\stackrel{(a)}{=} 3C^2 q^{2k}\left\|x_i^0\right\|^2 + \frac{3\gamma^2 C^2}{1-q}\sum_{s=0}^{k} q^{k-s}\mathbb{E}\left[\left\|\nabla f_i\left(z_i^s;\xi_i^s\right) - \nabla f_i\left(z_i^s\right) + \nabla f_i\left(z_i^s\right)\right\|^2\right] \\
&\quad + \frac{3\gamma^2 C^2 \beta K}{1-q}\sum_{s=0}^{k} q^{k-s}\mathbb{E}\left[\left\|\nabla f_i\left(z_i^s;\xi_i^s\right)\right\|^2\right] \\
&\stackrel{(b)}{=} 3C^2 q^{2k}\left\|x_i^0\right\|^2 + \frac{3\gamma^2 C^2}{1-q}\sum_{s=0}^{k} q^{k-s}\mathbb{E}\left[\left\|\nabla f_i\left(z_i^s;\xi_i^s\right) - \nabla f_i\left(z_i^s\right)\right\|^2\right] \\
&\quad + \frac{3\gamma^2 C^2}{1-q}\sum_{s=0}^{k} q^{k-s}\mathbb{E}\left[\left\|\nabla f_i\left(z_i^s\right)\right\|^2\right] + \frac{3\gamma^2 C^2 \beta K}{1-q}\sum_{s=0}^{k} q^{k-s}\mathbb{E}\left[\left\|\nabla f_i\left(z_i^s;\xi_i^s\right)\right\|^2\right],
\end{aligned}
\tag{29}
$$

where we used (6) in $(a)$, and (7) in $(b)$.

Further, using (8) and Lemma 2, the above inequality becomes

$$
\begin{aligned}
&\mathbb{E}\left[\left\|z_i^k - \bar{x}^k\right\|^2\right] \\
&\leqslant 3C^2 q^{2k}\left\|x_i^0\right\|^2 + \frac{3\gamma^2 C^2}{1-q}\sum_{s=0}^{k} q^{k-s}\cdot\zeta^2 + \frac{3\gamma^2 C^2 \beta K}{1-q}\sum_{s=0}^{k} q^{k-s}\mathbb{E}\left[\left\|\nabla f_i\left(z_i^s;\xi_i^s\right)\right\|^2\right] \\
&\quad + \frac{3\gamma^2 C^2}{1-q}\sum_{s=0}^{k} q^{k-s}\mathbb{E}\left[3L^2\left\|z_i^s - \bar{x}^s\right\|^2 + 3b^2 + 3\left\|\nabla f\left(\bar{x}^s\right)\right\|^2\right] \\
&\leqslant 3C^2 q^{2k}\left\|x_i^0\right\|^2 + \frac{3\gamma^2 C^2}{(1-q)^2}\left(\zeta^2 + 3b^2\right) + \frac{9\gamma^2 L^2 C^2}{1-q}\sum_{s=0}^{k} q^{k-s}\mathbb{E}\left[\left\|z_i^s - \bar{x}^s\right\|^2\right] \\
&\quad + \frac{9\gamma^2 C^2}{1-q}\sum_{s=0}^{k} q^{k-s}\mathbb{E}\left[\left\|\nabla f\left(\bar{x}^s\right)\right\|^2\right] + \frac{3\gamma^2 C^2 \beta K}{1-q}\sum_{s=0}^{k} q^{k-s}\mathbb{E}\left[\left\|\nabla f_i\left(z_i^s;\xi_i^s\right)\right\|^2\right].
\end{aligned}
\tag{30}
$$

Summing (30) from $i=1$ to $n$ and dividing by $n$, we have

$$
\begin{aligned}
&\frac{1}{n}\sum_{i=1}^{n}\mathbb{E}\left[\left\|z_i^k - \bar{x}^k\right\|^2\right] \\
&\leqslant \frac{3C^2 q^{2k}}{n}\sum_{i=1}^{n}\left\|x_i^0\right\|^2 + \frac{3\gamma^2 C^2}{(1-q)^2}\left(\zeta^2 + 3b^2\right) + \frac{9\gamma^2 L^2 C^2}{1-q}\sum_{s=0}^{k} q^{k-s}\cdot\frac{1}{n}\sum_{i=1}^{n}\mathbb{E}\left[\left\|z_i^s - \bar{x}^s\right\|^2\right] \\
&\quad + \frac{9\gamma^2 C^2}{1-q}\sum_{s=0}^{k} q^{k-s}\mathbb{E}\left[\left\|\nabla f\left(\bar{x}^s\right)\right\|^2\right] + \frac{3\gamma^2 C^2 \beta K}{1-q}\sum_{s=0}^{k} q^{k-s}\cdot\frac{1}{n}\sum_{i=1}^{n}\mathbb{E}\left[\left\|\nabla f_i\left(z_i^s;\xi_i^s\right)\right\|^2\right].
\end{aligned}
\tag{31}
$$

Summing (31) from $k = 0$ to $K - 1$, we obtain

$$\sum_{k=0}^{K-1} \cdot \frac{1}{n} \sum_{i=1}^{n} \mathbb{E}\left[\left\|z_i^k - \bar{x}^k\right\|^2\right]$$

$$\leqslant \frac{3C^2}{(1-q)^2 n} \sum_{i=1}^{n} \left\|x_i^0\right\|^2 + K \cdot \frac{3\gamma^2 C^2}{(1-q)^2} \left(\zeta^2 + 3b^2\right) + \frac{9\gamma^2 L^2 C^2}{1-q} \sum_{k=0}^{K-1} \sum_{s=0}^{k} q^{k-s} \cdot \frac{1}{n} \sum_{i=1}^{n} \mathbb{E}\left[\left\|z_i^s - \bar{x}^s\right\|^2\right]$$

$$+ \frac{9\gamma^2 C^2}{1-q} \sum_{k=0}^{K-1} \sum_{s=0}^{k} q^{k-s} \mathbb{E}\left[\left\|\nabla f\left(\bar{x}^s\right)\right\|^2\right] + \frac{3\gamma^2 C^2 \beta K}{1-q} \sum_{k=0}^{K-1} \sum_{s=0}^{k} q^{k-s} \cdot \frac{1}{n} \sum_{i=1}^{n} \mathbb{E}\left[\left\|\nabla f_i\left(z_i^s; \xi_i^s\right)\right\|^2\right]$$

$$\overset{(c)}{\leqslant} \frac{3C^2}{(1-q)^2 n} \sum_{i=1}^{n} \left\|x_i^0\right\|^2 + K \cdot \frac{3\gamma^2 C^2}{(1-q)^2} \left(\zeta^2 + 3b^2\right) + \frac{9\gamma^2 L^2 C^2}{(1-q)^2} \sum_{k=0}^{K-1} \frac{1}{n} \sum_{i=1}^{n} \mathbb{E}\left[\left\|z_i^k - \bar{x}^k\right\|^2\right]$$

$$+ \frac{9\gamma^2 C^2}{(1-q)^2} \sum_{k=0}^{K-1} \mathbb{E}\left[\left\|\nabla f\left(\bar{x}^k\right)\right\|^2\right] + \frac{3\gamma^2 C^2 \beta K}{(1-q)^2} \sum_{k=0}^{K-1} \frac{1}{n} \sum_{i=1}^{n} \mathbb{E}\left[\left\|\nabla f_i\left(z_i^k; \xi_i^k\right)\right\|^2\right].$$

$$(32)$$

where in $(c)$ we used the fact that $\sum_{k=0}^{K-1} \sum_{s=0}^{k} q^{k-s} a_s \leqslant \frac{1}{1-q} \sum_{k=0}^{K-1} a_k$.

Rearranging the term in (32), we obtain

$$\left(1 - \frac{9\gamma^2 L^2 C^2}{(1-q)^2}\right) \sum_{k=0}^{K-1} \cdot \frac{1}{n} \sum_{i=1}^{n} \mathbb{E}\left[\left\|z_i^k - \bar{x}^k\right\|^2\right]$$

$$\leqslant \frac{9\gamma^2 C^2}{(1-q)^2} \sum_{k=0}^{K-1} \mathbb{E}\left[\left\|\nabla f\left(\bar{x}^k\right)\right\|^2\right] + K \cdot \frac{3\gamma^2 C^2}{(1-q)^2} \left(\zeta^2 + 3b^2\right)$$

$$+ \frac{3C^2}{(1-q)^2 n} \sum_{i=1}^{n} \left\|x_i^0\right\|^2 + \frac{3\gamma^2 C^2 \beta K}{(1-q)^2} \sum_{k=0}^{K-1} \frac{1}{n} \sum_{i=1}^{n} \mathbb{E}\left[\left\|\nabla f_i\left(z_i^k; \xi_i^k\right)\right\|^2\right],$$

$$(33)$$

which completes the proof. $\qquad\square$

### A.3 PROOF OF THEOREM 1

With two supporting lemmas (Lemma 3 and Lemma 5) in the previous section, we are now ready to prove Theorem 1. We first upper bound the error term arising from injected DP noise as follows:

$$\beta K \sum_{k=0}^{K-1} \frac{1}{n} \sum_{i=1}^{n} \mathbb{E}\left[\left\|\nabla f_i(z_i^k; \xi_i^k)\right\|^2\right]$$

$$= \beta K \sum_{k=0}^{K-1} \frac{1}{n} \sum_{i=1}^{n} \mathbb{E}\left[\left\|\nabla f_i\left(z_i^k; \xi_i^k\right) - \nabla f_i\left(z_i^k\right) + \nabla f_i\left(z_i^k\right)\right\|^2\right]$$

$$= \beta K \sum_{k=0}^{K-1} \frac{1}{n} \sum_{i=1}^{n} \mathbb{E}\left[\left\|\nabla f_i\left(z_i^k; \xi_i^k\right) - \nabla f_i\left(z_i^k\right)\right\|^2\right] + \beta K \sum_{k=0}^{K-1} \frac{1}{n} \sum_{i=1}^{n} \mathbb{E}\left[\left\|\nabla f_i\left(z_i^k\right)\right\|^2\right] \quad (34)$$

$$\overset{(a)}{\leqslant} \beta K \sum_{k=0}^{K-1} \frac{1}{n} \sum_{i=1}^{n} \zeta^2 + \beta K \sum_{k=0}^{K-1} \frac{1}{n} \sum_{i=1}^{n} \mathbb{E}\left[3L^2 \left\|z_i^k - \bar{x}^k\right\|^2 + 3b^2 + 3\left\|\nabla f\left(\bar{x}^k\right)\right\|^2\right]$$

$$= \beta K^2 \left(\zeta^2 + 3b^2\right) + 3\beta K \sum_{k=0}^{K-1} \mathbb{E}\left[\left\|\nabla f\left(\bar{x}^k\right)\right\|^2\right] + 3L^2 \beta K \sum_{k=0}^{K-1} \frac{1}{n} \sum_{i=1}^{n} \mathbb{E}\left[\left\|z_i^k - \bar{x}^k\right\|^2\right],$$

where in $(a)$ we used (8) and Lemma 2.

Substituting (34) into (18), we have

$$
\frac{\gamma}{2} \sum_{k=0}^{K-1} \mathbb{E}\left[\left\|\nabla f\left(\bar{x}^k\right)\right\|^2\right] + \frac{\gamma\left(1-\gamma L\right)}{2} \sum_{k=0}^{K-1} \mathbb{E}\left[\left\|\frac{1}{n}\sum_{i=1}^{n}\nabla f_i\left(z_i^k\right)\right\|^2\right]
$$

$$
\leqslant f\left(\bar{x}^0\right) - f^* + \frac{\gamma L^2}{2} \sum_{k=0}^{K-1}\frac{1}{n}\sum_{i=1}^{n}\mathbb{E}\left[\left\|z_i^k - \bar{x}^k\right\|^2\right] + \frac{\gamma^2 LK}{2n}\zeta^2 + \frac{\gamma^2 L\beta K^2}{2n}\left(\zeta^2 + 3b^2\right) \tag{35}
$$

$$
+ \frac{3\gamma^2 L\beta K}{2n}\sum_{k=0}^{K-1}\mathbb{E}\left[\left\|\nabla f\left(\bar{x}^k\right)\right\|^2\right] + \frac{3\gamma^2 L^3\beta K}{2n}\sum_{k=0}^{K-1}\frac{1}{n}\sum_{i=1}^{n}\mathbb{E}\left[\left\|z_i^k - \bar{x}^k\right\|^2\right].
$$

Rearranging terms in the above inequality, we can obtain

$$
\left(\frac{\gamma}{2} - \frac{3\gamma^2 L\beta K}{2n}\right) \sum_{k=0}^{K-1} \mathbb{E}\left[\left\|\nabla f\left(\bar{x}^k\right)\right\|^2\right] + \frac{\gamma\left(1-\gamma L\right)}{2} \sum_{k=0}^{K-1} \mathbb{E}\left[\left\|\frac{1}{n}\sum_{i=1}^{n}\nabla f_i\left(z_i^k\right)\right\|^2\right]
$$

$$
\leqslant f\left(\bar{x}^0\right) - f^* + \left(\frac{\gamma L^2}{2} + \frac{3\gamma^2 L^3\beta K}{2n}\right) \sum_{k=0}^{K-1}\frac{1}{n}\sum_{i=1}^{n}\mathbb{E}\left[\left\|z_i^k - \bar{x}^k\right\|^2\right] + \frac{\gamma^2 LK}{2n}\zeta^2 \tag{36}
$$

$$
+ \frac{\gamma^2 L\beta K^2}{2n}\left(\zeta^2 + 3b^2\right).
$$

Similarly, substituting (34) into (26), we have

$$
\left(1 - \frac{9\gamma^2 L^2 C^2}{\left(1-q\right)^2}\right) \sum_{k=0}^{K-1}\cdot\frac{1}{n}\sum_{i=1}^{n}\mathbb{E}\left[\left\|z_i^k - \bar{x}^k\right\|^2\right]
$$

$$
\leqslant \frac{9\gamma^2 C^2}{\left(1-q\right)^2}\sum_{k=0}^{K-1}\mathbb{E}\left[\left\|\nabla f\left(\bar{x}^k\right)\right\|^2\right] + \left(1+\beta K\right)K\cdot\frac{3\gamma^2 C^2}{\left(1-q\right)^2}\left(\zeta^2 + 3b^2\right) + \frac{3C^2}{\left(1-q\right)^2 n}\sum_{i=1}^{n}\left\|x_i^0\right\|^2
$$

$$
+ \frac{9\gamma^2 C^2\beta K}{\left(1-q\right)^2}\sum_{k=0}^{K-1}\mathbb{E}\left[\left\|\nabla f\left(\bar{x}^k\right)\right\|^2\right] + \frac{9\gamma^2 L^2 C^2\beta K}{\left(1-q\right)^2}\sum_{k=0}^{K-1}\frac{1}{n}\sum_{i=1}^{n}\mathbb{E}\left[\left\|z_i^k - \bar{x}^k\right\|^2\right].
$$

Rearranging terms in the above inequality, we can get

$$
\left(1 - \left(1+\beta K\right)\cdot\frac{9\gamma^2 L^2 C^2}{\left(1-q\right)^2}\right) \sum_{k=0}^{K-1}\frac{1}{n}\sum_{i=1}^{n}\mathbb{E}\left[\left\|z_i^k - \bar{x}^k\right\|^2\right]
$$

$$
\leqslant \left(1+\beta K\right)\cdot\frac{9\gamma^2 C^2}{\left(1-q\right)^2}\sum_{k=0}^{K-1}\mathbb{E}\left[\left\|\nabla f\left(\bar{x}^k\right)\right\|^2\right] + \left(1+\beta K\right)K\cdot\frac{3\gamma^2 C^2}{\left(1-q\right)^2}\left(\zeta^2 + 3b^2\right) \tag{37}
$$

$$
+ \frac{3C^2}{\left(1-q\right)^2 n}\sum_{i=1}^{n}\left\|x_i^0\right\|^2.
$$

Substituting $K = \frac{1}{\beta}\stackrel{(6)}{=}\frac{J^2\epsilon^2}{dc_2^2\log\left(\frac{1}{\delta}\right)}$ into (36) and (37), those two inequalities become now

$$
\left(\frac{1}{2} - \frac{3\gamma L}{2n}\right)\cdot\frac{1}{K}\sum_{k=0}^{K-1}\mathbb{E}\left[\left\|\nabla f\left(\bar{x}^k\right)\right\|^2\right] + \frac{1-\gamma L}{2K}\sum_{k=0}^{K-1}\mathbb{E}\left[\left\|\frac{1}{n}\sum_{i=1}^{n}\nabla f_i\left(z_i^k\right)\right\|^2\right]
$$

$$
\leqslant \frac{f\left(\bar{x}^0\right) - f^*}{\gamma K} + \left(\frac{L^2}{2} + \frac{3\gamma L^3}{2n}\right)\cdot\frac{1}{K}\sum_{k=0}^{K-1}\frac{1}{n}\sum_{i=1}^{n}\mathbb{E}\left[\left\|z_i^k - \bar{x}^k\right\|^2\right] + \frac{\gamma L}{2n}\left(2\zeta^2 + 3b^2\right) \tag{38}
$$

and

$$\left(1 - \frac{18\gamma^2 L^2 C^2}{(1-q)^2}\right) \cdot \frac{1}{K} \sum_{k=0}^{K-1} \frac{1}{n} \sum_{i=1}^n \mathbb{E}\left[\left\|z_i^k - \bar{x}^k\right\|^2\right]$$
$$\leqslant \frac{18\gamma^2 C^2}{(1-q)^2} \cdot \frac{1}{K} \sum_{k=0}^{K-1} \mathbb{E}\left[\left\|\nabla f\left(\bar{x}^k\right)\right\|^2\right] + \frac{6\gamma^2 C^2 \left(\zeta^2 + 3b^2\right)}{(1-q)^2} + \frac{3C^2}{(1-q)^2 nK} \sum_{i=1}^n \left\|x_i^0\right\|^2. \tag{39}$$

If $\gamma$ satisfies

$$\gamma \leqslant \frac{1-q}{6LC}, \tag{40}$$

(39) becomes

$$\frac{1}{2K} \sum_{k=0}^{K-1} \frac{1}{n} \sum_{i=1}^n \mathbb{E}\left[\left\|z_i^k - \bar{x}^k\right\|^2\right] \leqslant \frac{18\gamma^2 C^2}{(1-q)^2} \cdot \frac{1}{K} \sum_{k=0}^{K-1} \mathbb{E}\left[\left\|\nabla f\left(\bar{x}^k\right)\right\|^2\right]$$
$$+ \frac{6\gamma^2 C^2 \left(\zeta^2 + 3b^2\right)}{(1-q)^2} + \frac{3C^2}{(1-q)^2 nK} \sum_{i=1}^n \left\|x_i^0\right\|^2. \tag{41}$$

Substituting (41) into (38), we have

$$\left(\frac{1}{2} - \frac{3\gamma L}{2n} - \frac{18\gamma^2 L^2 C^2}{(1-q)^2} - \frac{54\gamma^3 L^3 C^2}{(1-q)^2 n}\right) \cdot \frac{1}{K} \sum_{k=0}^{K-1} \mathbb{E}\left[\left\|\nabla f\left(\bar{x}^k\right)\right\|^2\right]$$
$$+ \frac{1-\gamma L}{2} \cdot \frac{1}{K} \sum_{k=0}^{K-1} \mathbb{E}\left[\left\|\frac{1}{n} \sum_{i=1}^n \nabla f_i\left(z_i^k\right)\right\|^2\right]$$
$$\leqslant \frac{f\left(\bar{x}^0\right) - f^*}{\gamma K} + \left(L^2 + \frac{3\gamma L^3}{n}\right) \left(\frac{6\gamma^2 C^2 \left(\zeta^2 + 3b^2\right)}{(1-q)^2} + \frac{3C^2}{(1-q)^2 nK} \sum_{i=1}^n \left\|x_i^0\right\|^2\right)$$
$$+ \frac{\gamma L}{2n} \left(2\zeta^2 + 3b^2\right). \tag{42}$$

If $\gamma$ further satisfies

$$\gamma \leqslant \min\left\{\frac{(1-q)^2}{24nLC^2}, \frac{n}{12L}, \frac{1}{L}\right\}, \tag{43}$$

for (42), we have

$$\frac{1}{4} \cdot \frac{1}{K} \sum_{k=0}^{K-1} \mathbb{E}\left[\left\|\nabla f\left(\bar{x}^k\right)\right\|^2\right]$$
$$\leqslant \left(\frac{1}{2} - \frac{3\gamma L}{2n} - \frac{18\gamma^2 L^2 C^2}{(1-q)^2} - \frac{54\gamma^3 L^3 C^2}{(1-q)^2 n}\right) \cdot \frac{1}{K} \sum_{k=0}^{K-1} \mathbb{E}\left[\left\|\nabla f\left(\bar{x}^k\right)\right\|^2\right]$$
$$+ \frac{1-\gamma L}{2} \cdot \frac{1}{K} \sum_{k=0}^{K-1} \mathbb{E}\left[\left\|\frac{1}{n} \sum_{i=1}^n \nabla f_i\left(z_i^k\right)\right\|^2\right]$$
$$\leqslant \frac{f\left(\bar{x}^0\right) - f^*}{\gamma K} + \frac{1}{K} \cdot \frac{6L^2 C^2}{(1-q)^2 n} \sum_{i=1}^n \left\|x_i^0\right\|^2 + \frac{12\gamma^2 L^2 C^2 \left(\zeta^2 + 3b^2\right)}{(1-q)^2}$$
$$+ \frac{\gamma L}{2n} \left(2\zeta^2 + 3b^2\right), \tag{44}$$

i.e.,

$$\frac{1}{K} \sum_{k=0}^{K-1} \mathbb{E}\left[\left\|\nabla f\left(\bar{x}^k\right)\right\|^2\right] \leqslant \frac{4\left(f\left(\bar{x}^0\right) - f^*\right)}{\gamma K} + \frac{1}{K} \cdot \frac{24L^2 C^2}{(1-q)^2 n} \sum_{i=1}^n \left\|x_i^0\right\|^2$$
$$+ \frac{48\gamma^2 L^2 C^2 \left(\zeta^2 + 3b^2\right)}{(1-q)^2} + \frac{2\gamma L}{n} \left(2\zeta^2 + 3b^2\right). \tag{45}$$

By now, the step size $\gamma$ need to satisfy (40) and (43), i.e.,

$$\gamma \leqslant \underbrace{\min\left\{\frac{(1-q)^2}{24nLC^2}, \frac{1-q}{6LC}, \frac{n}{12L}, \frac{1}{L}\right\}}_{\triangleq \hat{\gamma}(C,q)}. \tag{46}$$

Now we set the step size $\gamma$ as

$$\gamma = \frac{1}{\sqrt{K/n} + \hat{\gamma}(C,q)^{-1}} = \frac{1}{\sqrt{\frac{1}{n\beta}} + \hat{\gamma}(C,q)^{-1}} \overset{(6)}{=} \frac{1}{\frac{J\epsilon}{c_2\sqrt{nd\log\left(\frac{1}{\delta}\right)}} + \hat{\gamma}(C,q)^{-1}}, \tag{47}$$

then (45) can be further bounded as

$$
\begin{aligned}
&\frac{1}{K}\sum_{k=0}^{K-1}\mathbb{E}\left[\left\|\nabla f\left(\bar{x}^k\right)\right\|^2\right] \\
&\leqslant \frac{4\left(f\left(\bar{x}^0\right) - f^*\right)}{\gamma K} + \frac{1}{K}\cdot\frac{24L^2C^2}{(1-q)^2 n}\sum_{i=1}^{n}\left\|x_i^0\right\|^2 \\
&\quad + \frac{48\gamma^2L^2C^2\left(\zeta^2 + 3b^2\right)}{(1-q)^2} + \frac{2\gamma L}{n}\left(2\zeta^2 + 3b^2\right) \\
&\overset{(47)}{\leqslant} \frac{4\left(f\left(\bar{x}^0\right) - f^*\right)}{\sqrt{nK}} + \frac{2L\left(2\zeta^2 + 3b^2\right)}{\sqrt{nK}} + \frac{4\left(f\left(\bar{x}^0\right) - f^*\right)}{\hat{\gamma}(C,q)K} \\
&\quad + \frac{1}{K}\cdot\frac{24L^2C^2}{(1-q)^2 n}\sum_{i=1}^{n}\left\|x_i^0\right\|^2 + \frac{1}{K}\cdot\frac{48nL^2C^2\left(\zeta^2 + 3b^2\right)}{(1-q)^2} \\
&= \frac{4\left(f\left(\bar{x}^0\right) - f^*\right) + 2L\left(2\zeta^2 + 3b^2\right)}{\sqrt{nK}} \\
&\quad + \frac{1}{K}\left(\frac{24L^2C^2}{(1-q)^2 n}\sum_{i=1}^{n}\left\|x_i^0\right\|^2 + \frac{48nL^2C^2\left(\zeta^2 + 3b^2\right)}{(1-q)^2} + \frac{4\left(f\left(\bar{x}^0\right) - f^*\right)}{\hat{\gamma}(C,q)}\right).
\end{aligned} \tag{48}
$$

Knowing that $K$ is chosen as

$$K = \frac{1}{\beta} = \frac{J^2\epsilon^2}{dc_2^2\log\left(\frac{1}{\delta}\right)}, \tag{49}$$

(48) becomes

$$
\begin{aligned}
&\frac{1}{K}\sum_{k=0}^{K-1}\mathbb{E}\left[\left\|\nabla f\left(\bar{x}^k\right)\right\|^2\right] \\
&\overset{(46)}{\leqslant} \frac{c_2\sqrt{d\log\left(\frac{1}{\delta}\right)}}{\sqrt{n}J\epsilon}\cdot\left[4\left(f\left(\bar{x}^0\right) - f^*\right) + 2L\left(2\zeta^2 + 3b^2\right)\right] \\
&\quad + \frac{c_2^2 d\log\left(\frac{1}{\delta}\right)}{J^2\epsilon^2}\cdot\left[\frac{24L^2C^2}{(1-q)^2 n}\sum_{i=1}^{n}\left\|x_i^0\right\|^2 + \frac{48nL^2C^2\left(\zeta^2 + 3b^2\right)}{(1-q)^2}\right] \\
&\quad + \frac{c_2^2 d\log\left(\frac{1}{\delta}\right)}{J^2\epsilon^2}\cdot 4\left(f\left(\bar{x}^0\right) - f^*\right)\cdot\max\left\{\frac{24nLC^2}{(1-q)^2}, \frac{6LC}{1-q}, \frac{12L}{n}, L\right\}.
\end{aligned} \tag{50}
$$

According to the mild assumption

$$J \geqslant \frac{n^{\frac{3}{2}}c_2\sqrt{d\log\left(\frac{1}{\delta}\right)}}{\epsilon}, \tag{51}$$

(50) can be further bounded as

$$\frac{1}{K} \sum_{k=0}^{K-1} \mathbb{E}\left[\left\|\nabla f\left(\bar{x}^k\right)\right\|^2\right]$$

$$\leqslant \frac{c_2 \sqrt{d \log\left(\frac{1}{\delta}\right)}}{\sqrt{n} J \epsilon} \cdot \left[4\left(f\left(\bar{x}^0\right) - f^*\right) + 2L\left(2\zeta^2 + 3b^2\right)\right]$$

$$+ \frac{c_2 \sqrt{d \log\left(\frac{1}{\delta}\right)}}{J \epsilon} \cdot \frac{1}{n^{3/2}} \cdot \left[\frac{24 L^2 C^2}{(1-q)^2 n} \sum_{i=1}^n \left\|x_i^0\right\|^2 + \frac{48 n L^2 C^2 \left(\zeta^2 + 3b^2\right)}{(1-q)^2}\right]$$

$$+ \frac{c_2 \sqrt{d \log\left(\frac{1}{\delta}\right)}}{J \epsilon} \cdot \frac{1}{n^{3/2}} \cdot 4\left(f\left(\bar{x}^0\right) - f^*\right) \cdot \max\left\{\frac{24 n L C^2}{(1-q)^2}, \frac{6 L C}{1-q}, \frac{12 L}{n}, L\right\}$$

$$\leqslant \frac{c_2 \sqrt{d \log\left(\frac{1}{\delta}\right)}}{\sqrt{n} J \epsilon} \cdot \left[4\left(f\left(\bar{x}^0\right) - f^*\right) + 2L\left(2\zeta^2 + 3b^2\right)\right]$$

$$+ \frac{c_2 \sqrt{d \log\left(\frac{1}{\delta}\right)}}{\sqrt{n} J \epsilon} \cdot \left[\frac{24 L^2 C^2}{(1-q)^2 n} \sum_{i=1}^n \left\|x_i^0\right\|^2 + \frac{48 L^2 C^2 \left(\zeta^2 + 3b^2\right)}{(1-q)^2}\right]$$

$$+ \frac{c_2 \sqrt{d \log\left(\frac{1}{\delta}\right)}}{\sqrt{n} J \epsilon} \cdot 4\left(f\left(\bar{x}^0\right) - f^*\right) \cdot \max\left\{\frac{24 L C^2}{(1-q)^2}, \frac{6 L C}{1-q}, 13 L\right\}$$

$$= \mathcal{O}\left(\frac{\sqrt{d \log\left(\frac{1}{\delta}\right)}}{\sqrt{n} J \epsilon}\right),$$

which completes the proof of Theorem 1.

### A.4 PROOF OF PROPOSITION 3

Now we provide the complete proof of Proposition 3.

If the step size $\gamma$ satisfies

$$\gamma \leqslant \frac{1-q}{\sqrt{18} L C}, \tag{52}$$

(26) becomes

$$\frac{1}{2} \sum_{k=0}^{K-1} \frac{1}{n} \sum_{i=1}^n \mathbb{E}\left[\left\|z_i^k - \bar{x}^k\right\|^2\right]$$

$$\leqslant \frac{9 \gamma^2 C^2}{(1-q)^2} \sum_{k=0}^{K-1} \mathbb{E}\left[\left\|\nabla f\left(\bar{x}^k\right)\right\|^2\right] + K \cdot \frac{3 \gamma^2 C^2}{(1-q)^2}\left(\zeta^2 + 3b^2\right) \tag{53}$$

$$+ \frac{3 C^2}{(1-q)^2 n} \sum_{i=1}^n \left\|x_i^0\right\|^2 + \frac{3 \gamma^2 C^2 \beta K}{(1-q)^2} \sum_{k=0}^{K-1} \frac{1}{n} \sum_{i=1}^n \mathbb{E}\left[\left\|\nabla f_i\left(z_i^k; \xi_i^k\right)\right\|^2\right].$$

Substituting (53) into (18), we obtain

$$\left(\frac{\gamma}{2} - \frac{9 \gamma^3 L^2 C^2}{(1-q)^2}\right) \sum_{k=0}^{K-1} \mathbb{E}\left[\left\|\nabla f\left(\bar{x}^k\right)\right\|^2\right] + \frac{\gamma(1-\gamma L)}{2} \sum_{k=0}^{K-1} \mathbb{E}\left[\left\|\frac{1}{n} \sum_{i=1}^n \nabla f_i\left(z_i^k\right)\right\|^2\right]$$

$$\leqslant f\left(\bar{x}^0\right) - f^* + \frac{\gamma^2 L K}{2n} \zeta^2 + K \cdot \frac{3 \gamma^3 L^2 C^2}{(1-q)^2}\left(\zeta^2 + 3b^2\right) + \frac{3 \gamma L^2 C^2}{(1-q)^2 n} \sum_{i=1}^n \left\|x_i^0\right\|^2 \tag{54}$$

$$+ \left(\frac{3 \gamma^3 L^2 C^2 \beta K}{(1-q)^2} + \frac{\gamma^2 L \beta K}{2n}\right) \sum_{k=0}^{K-1} \frac{1}{n} \sum_{i=1}^n \mathbb{E}\left[\left\|\nabla f_i\left(z_i^k; \xi_i^k\right)\right\|^2\right].$$

Dividing by $\gamma K$ on both sides of the above inequality yields

$$
\left( \frac{1}{2} - \frac{9\gamma^2 L^2 C^2}{(1-q)^2} \right) \frac{1}{K} \sum_{k=0}^{K-1} \mathbb{E} \left[ \left\| \nabla f \left( \bar{x}^k \right) \right\|^2 \right] + \frac{1-\gamma L}{2K} \sum_{k=0}^{K-1} \mathbb{E} \left[ \left\| \frac{1}{n} \sum_{i=1}^{n} \nabla f_i \left( z_i^k \right) \right\|^2 \right]
$$

$$
\leqslant \frac{f \left( \bar{x}^0 \right) - f^*}{\gamma K} + \frac{\gamma L}{2n} \zeta^2 + \frac{3\gamma^2 L^2 C^2}{(1-q)^2} \left( \zeta^2 + 3b^2 \right) + \frac{3 L^2 C^2}{(1-q)^2 nK} \sum_{i=1}^{n} \left\| x_i^0 \right\|^2 \tag{55}
$$

$$
+ \left( \frac{3\gamma^2 L^2 C^2 \beta}{(1-q)^2} + \frac{\gamma L \beta}{2n} \right) \sum_{k=0}^{K-1} \frac{1}{n} \sum_{i=1}^{n} \mathbb{E} \left[ \left\| \nabla f_i \left( z_i^k; \xi_i^k \right) \right\|^2 \right].
$$

If the step size $\gamma$ satisfies

$$
\gamma \leqslant \min \left\{ \frac{1-q}{6LC}, \frac{1}{L} \right\}, \tag{56}
$$

(55) becomes now

$$
\frac{1}{4} \cdot \frac{1}{K} \sum_{k=0}^{K-1} \mathbb{E} \left[ \left\| \nabla f \left( \bar{x}^k \right) \right\|^2 \right]
$$

$$
\leqslant \frac{f \left( \bar{x}^0 \right) - f^*}{\gamma K} + \frac{\gamma L}{2n} \zeta^2 + \frac{3\gamma^2 L^2 C^2}{(1-q)^2} \left( \zeta^2 + 3b^2 \right) + \frac{3 L^2 C^2}{(1-q)^2 nK} \sum_{i=1}^{n} \left\| x_i^0 \right\|^2 \tag{57}
$$

$$
+ \left( \frac{3\gamma^2 L^2 C^2 \beta}{(1-q)^2} + \frac{\gamma L \beta}{2n} \right) \sum_{k=0}^{K-1} \frac{1}{n} \sum_{i=1}^{n} \mathbb{E} \left[ \left\| \nabla f_i \left( z_i^k; \xi_i^k \right) \right\|^2 \right].
$$

By now, the step size $\gamma$ needs to satisfy (52) and (56), i.e.,

$$
\gamma \leqslant \min \left\{ \frac{1-q}{6LC}, \frac{1}{L} \right\}. \tag{58}
$$

We thus complete the proof of Proposition 3.

### A.5 PROOF OF PROPOSITION 4

The proof of Proposition 4 shares the similarity with that of Proposition 3, except for the processing of privacy noise-related terms.

According to (83) in Algorithm 2 and (5) in Theorem 2, we have the variance of injected Gaussian noise for each node $i$ at each iteration $k$ as follows

$$
\mathbb{E} \left[ \left\| N_i^k \right\|^2 \right] = d\sigma^2 G^2 = \underbrace{\frac{dc_2^2 \log \left( \frac{1}{\delta} \right)}{J^2 \epsilon^2}}_{\beta} \cdot K G^2. \tag{59}
$$

Therefore, $A_3$ in (22) becomes

$$
A_3 = \mathbb{E} \left[ \left\| \frac{1}{n} \sum_{i=1}^{n} N_i^k \right\|^2 \right] = \frac{1}{n^2} \sum_{i=1}^{n} \mathbb{E} \left[ \left\| N_i^k \right\|^2 \right] \overset{(59)}{=} \frac{\beta K}{n^2} \sum_{i=1}^{n} G^2. \tag{60}
$$

Following the proof of Lemma 3 with the above new $A_3$, we have

$$
\frac{\gamma}{2} \sum_{k=0}^{K-1} \mathbb{E} \left[ \left\| \nabla f \left( \bar{x}^k \right) \right\|^2 \right] + \frac{\gamma (1-\gamma L)}{2} \sum_{k=0}^{K-1} \mathbb{E} \left[ \left\| \frac{1}{n} \sum_{i=1}^{n} \nabla f_i \left( z_i^k \right) \right\|^2 \right]
$$

$$
\leqslant f \left( \bar{x}^0 \right) - f^* + \frac{\gamma L^2}{2} \sum_{k=0}^{K-1} \frac{1}{n} \sum_{i=1}^{n} \mathbb{E} \left[ \left\| z_i^k - \bar{x}^k \right\|^2 \right] + \frac{\gamma^2 LK}{2n} \zeta^2 + \frac{\gamma^2 L\beta K}{2n} \sum_{k=0}^{K-1} \frac{1}{n} \sum_{i=1}^{n} G^2. \tag{61}
$$

Following the proof of Lemma 5 with the new noise variance $\mathbb{E}\left[\left\|N_i^k\right\|^2\right]$ (c.f., (59)), we have

$$
\left(1 - \frac{9\gamma^2 L^2 C^2}{(1-q)^2}\right) \sum_{k=0}^{K-1} \cdot \frac{1}{n} \sum_{i=1}^{n} \mathbb{E}\left[\left\|z_i^k - \bar{x}^k\right\|^2\right]
$$

$$
\leqslant \frac{9\gamma^2 C^2}{(1-q)^2} \sum_{k=0}^{K-1} \mathbb{E}\left[\left\|\nabla f\left(\bar{x}^k\right)\right\|^2\right] + K \cdot \frac{3\gamma^2 C^2}{(1-q)^2}\left(\zeta^2 + 3b^2\right) \tag{62}
$$

$$
+ \frac{3C^2}{(1-q)^2 n} \sum_{i=1}^{n}\left\|x_i^0\right\|^2 + \frac{3\gamma^2 C^2 \beta K}{(1-q)^2} \sum_{k=0}^{K-1} \frac{1}{n} \sum_{i=1}^{n} G^2.
$$

With (61) and (62) being the alternatives of (18) and (26) respectively, following the proof of Proposition 3 (c.f., (52)-(58)), we can easily obtain

$$
\frac{1}{4} \cdot \frac{1}{K} \sum_{k=0}^{K-1} \mathbb{E}\left[\left\|\nabla f\left(\bar{x}^k\right)\right\|^2\right]
$$

$$
\leqslant \frac{f\left(\bar{x}^0\right) - f^*}{\gamma K} + \frac{\gamma L}{2n}\zeta^2 + \frac{3\gamma^2 L^2 C^2}{(1-q)^2}\left(\zeta^2 + 3b^2\right) + \frac{3L^2 C^2}{(1-q)^2 nK} \sum_{i=1}^{n}\left\|x_i^0\right\|^2 \tag{63}
$$

$$
+ \left(\frac{3\gamma^2 L^2 C^2 \beta}{(1-q)^2} + \frac{\gamma L \beta}{2n}\right) \sum_{k=0}^{K-1} \frac{1}{n} \sum_{i=1}^{n} G^2,
$$

and the step size $\gamma$ need to satisfy

$$
\gamma \leqslant \min\left\{\frac{1-q}{6LC}, \frac{1}{L}\right\}. \tag{64}
$$

We thus complete the proof of Proposition 4.

## A.6 Derivation of utility bound for baseline centralized DP-SGD

We first make the following blanket assumptions for our theoretical analysis of centralized DP-SGD.

**Assumption 7** (*L*-smooth). *For any model parameter $x$ and $y$, we have*

$$
\left\|\nabla f\left(x\right) - \nabla f\left(y\right)\right\| \leqslant L\left\|x - y\right\|. \tag{65}
$$

**Assumption 8** (Unbiased gradient). *For any model parameter $x^k$, we have*

$$
\mathbb{E}\left[\nabla f\left(x^k; \xi^k\right)\right] = \nabla f\left(x^k\right). \tag{66}
$$

**Assumption 9** (Bounded variance). *For any model parameter $x^k$, we have*

$$
\mathbb{E}\left[\left\|\nabla f\left(x^k; \xi^k\right) - \nabla f\left(x^k\right)\right\|^2\right] \leqslant \zeta^2. \tag{67}
$$

**Assumption 10** (Bounded gradient). *For any $x \in \mathbb{R}^d$ and $\xi \in \{1, 2, ..., J\}$, there exists finite positive constant $G$ such that*

$$
\left\|\nabla f\left(x; \xi\right)\right\| \leqslant G. \tag{68}
$$

The update of centralized differentially private SGD is:

$$
x^{k+1} = x^k - \gamma\left(\nabla f\left(x^k; \xi^k\right) + n^k\right), \tag{69}
$$

where the randomized noise $n^k$ is drawn from the Gaussian distribution

$$
n^k \sim \mathcal{N}\left(0, \sigma^2 G^2 \mathbb{I}_d\right), \tag{70}
$$

and $\sigma$ is defined in Proposition 2.

With the above, we have

$$\mathbb{E}\left[\left\|n^k\right\|^2\right] = d \cdot \sigma^2 G^2 = \underbrace{\frac{dc_2^2 \log\left(\frac{1}{\delta}\right)}{J^2 \epsilon^2}}_{\beta} \cdot G^2 K. \tag{71}$$

Applying the descent lemma to $f$ at $x^k$ and $x^{k+1}$, we have

$$
\begin{aligned}
f\left(x^{k+1}\right) &\overset{(65)}{\leqslant} f\left(x^k\right) + \left\langle \nabla f\left(x^k\right), x^{k+1} - x^k \right\rangle + \frac{L}{2}\left\|x^{k+1} - x^k\right\|^2 \\
&\overset{(69)}{=} f\left(x^k\right) - \gamma \left\langle \nabla f\left(x^k\right), \nabla f\left(x^k;\xi^k\right) + n^k \right\rangle + \frac{\gamma^2 L}{2}\left\|\nabla f\left(x^k;\xi^k\right) + n^k\right\|^2 \\
&= f\left(x^k\right) - \gamma \left\langle \nabla f\left(x^k\right), \nabla f\left(x^k;\xi^k\right) + n^k \right\rangle + \frac{\gamma^2 L}{2}\left\|\nabla f\left(x^k;\xi^k\right)\right\|^2 \\
&\quad + \frac{\gamma^2 L}{2}\left\|n^k\right\|^2 + \gamma^2 L \left\langle \nabla f\left(x^k;\xi^k\right), n^k \right\rangle
\end{aligned} \tag{72}
$$

Taking expectation of both sides of the above inequality conditioned on $x^k$, we have

$$
\begin{aligned}
&\mathbb{E}\left[f\left(x^{k+1}\right)\big|x^k\right] \\
&\leqslant f\left(x^k\right) - \gamma \left\|\nabla f\left(x^k\right)\right\|^2 + \frac{\gamma^2 L}{2}\mathbb{E}\left[\left\|\nabla f\left(x^k;\xi^k\right)\right\|^2 \big|x^k\right] + \frac{\gamma^2 L}{2}\mathbb{E}\left[\left\|n^k\right\|^2 \big|x^k\right] \\
&\overset{(a)}{=} f\left(x^k\right) - \gamma \left\|\nabla f\left(x^k\right)\right\|^2 + \frac{\gamma^2 L}{2}\mathbb{E}\left[\left\|\nabla f\left(x^k;\xi^k\right) - \nabla f\left(x^k\right) + \nabla f\left(x^k\right)\right\|^2 \big|x^k\right] \\
&\quad + \frac{\gamma^2 L}{2}\cdot \beta G^2 K \\
&= f\left(x^k\right) - \gamma \left\|\nabla f\left(x^k\right)\right\|^2 + \frac{\gamma^2 L}{2}\mathbb{E}\left[\left\|\nabla f\left(x^k;\xi^k\right) - \nabla f\left(x^k\right)\right\|^2 \big|x^k\right] \\
&\quad + \frac{\gamma^2 L}{2}\left\|\nabla f\left(x^k\right)\right\|^2 + \frac{\gamma^2 L}{2}\cdot \beta G^2 K \\
&\overset{(b)}{\leqslant} f\left(x^k\right) - \gamma \left\|\nabla f\left(x^k\right)\right\|^2 + \frac{\gamma^2 L}{2}\left\|\nabla f\left(x^k\right)\right\|^2 + \frac{\gamma^2 L}{2}\zeta^2 + \frac{\gamma^2 L}{2}\cdot \beta G^2 K.
\end{aligned} \tag{73}
$$

where we used (71) in $(a)$ and (67) in $(b)$.

Taking total expectation on both sides of the above inequality, we have

$$\mathbb{E}\left[f\left(x^{k+1}\right)\right] \leqslant \mathbb{E}\left[f\left(x^k\right)\right] - \gamma\mathbb{E}\left[\left\|\nabla f\left(x^k\right)\right\|^2\right] + \frac{\gamma^2 L}{2}\mathbb{E}\left[\left\|\nabla f\left(x^k\right)\right\|^2\right] + \frac{\gamma^2 L}{2}\zeta^2 + \frac{\gamma^2 L}{2}\cdot \beta G^2 K. \tag{74}$$

Summing (74) from $k = 0$ to $K - 1$, we have

$$\left(\gamma - \frac{\gamma^2 L}{2}\right)\sum_{k=0}^{K-1}\mathbb{E}\left[\left\|\nabla f\left(x^k\right)\right\|^2\right] \leqslant f\left(x^0\right) - f^* + \frac{\gamma^2 L}{2}\zeta^2 \cdot K + \frac{\gamma^2 L}{2}\cdot \beta G^2 K^2. \tag{75}$$

Dividing $\gamma K$ on both sides, we have

$$\left(1 - \frac{\gamma L}{2}\right)\cdot \frac{1}{K}\sum_{k=0}^{K-1}\mathbb{E}\left[\left\|\nabla f\left(x^k\right)\right\|^2\right] \leqslant \frac{f\left(x^0\right) - f^*}{\gamma K} + \frac{\gamma L}{2}\zeta^2 + \frac{LG^2\beta}{2}\cdot \gamma K. \tag{76}$$

If $\gamma$ satisfies

$$\gamma \leqslant \frac{1}{L}, \tag{77}$$

(76) becomes

$$\frac{1}{K}\sum_{k=0}^{K-1}\mathbb{E}\left[\left\|\nabla f\left(x^k\right)\right\|^2\right] \leqslant \frac{2\left(f\left(x^0\right) - f^*\right)}{\gamma K} + \gamma L\zeta^2 + LG^2\beta \cdot \gamma K. \tag{78}$$

Now we set $\gamma$ as

$$\gamma = \frac{1}{\sqrt{K} + L}, \tag{79}$$

then (78) becomes

$$
\begin{aligned}
\frac{1}{K} \sum_{k=0}^{K-1} \mathbb{E}\left[\left\|\nabla f\left(x^k\right)\right\|^2\right] &\leqslant \frac{2\left(f\left(x^0\right) - f^*\right)}{\sqrt{K}} + \frac{2L\left(f\left(x^0\right) - f^*\right)}{K} + \frac{L\zeta^2}{\sqrt{K}} + LG^2\beta\sqrt{K} \\
&\leqslant \frac{2(1+L)\left(f\left(x^0\right) - f^*\right) + L\zeta^2}{\sqrt{K}} + LG^2\beta\sqrt{K}.
\end{aligned}
\tag{80}
$$

Regarding the right hand side of (80) as a function of $K$, we can obtain the optimal value of $K$ by minimizing this function, and the optimal value of $K$ is

$$K = \frac{2(1+L)\left(f\left(x^0\right) - f^*\right) + L\zeta^2}{LG^2\beta} \overset{(71)}{=} \frac{2(1+L)\left(f\left(x^0\right) - f^*\right) + L\zeta^2}{LG^2} \cdot \frac{J^2\epsilon^2}{dc_2^2 \log\left(\frac{1}{\delta}\right)}. \tag{81}$$

and the utility (80) becomes

$$
\begin{aligned}
\frac{1}{K} \sum_{k=0}^{K-1} \mathbb{E}\left[\left\|\nabla f\left(x^k\right)\right\|^2\right] &\leqslant 2G\sqrt{2(1+L)L\left(f\left(x^0\right) - f^*\right) + L^2\zeta^2} \cdot \sqrt{\beta} \\
&= 2c_2 G\sqrt{\left(2(1+L)L\left(f\left(x^0\right) - f^*\right) + L^2\zeta^2\right)} \cdot \frac{\sqrt{d\log\left(\frac{1}{\delta}\right)}}{J\epsilon} \\
&= \mathcal{O}\left(\frac{\sqrt{d\log\left(\frac{1}{\delta}\right)}}{J\epsilon}\right).
\end{aligned}
\tag{82}
$$

## B  MISSING PSEUDOCODES OF ALGORITHMS

---

**Algorithm 2** Differentially Private Decentralized Learning with Constant Gaussian Noise (**ConstD$^2$P**)

---

1: **Initialization:** $x_i^0 = z_i^0 \in \mathbb{R}^d$, $w_i^0 = 1$, learning rate $\gamma > 0$, total number of iterations $K$ and privacy budget $(\epsilon, \delta)$.
2: **for** $k = 0, 1, 2, ..., K - 1$, at node $i$, **do**
3:     Randomly samples a local training data $\xi_i^k$ with the sampling probability $\frac{1}{J}$;
4:     Computes stochastic gradient at $z_i^k$: $\nabla f_i(z_i^k; \xi_i^k)$;
5:     Draws randomized noise $N_i^k$ from the Gaussian distribution:

$$N_i^k \sim \mathcal{N}\left(0, \sigma^2 G^2 \mathbb{I}_d\right), \tag{83}$$

    where $\sigma$ is defined in Proposition 2, and $G$ is defined at (12);
6:     Differentially private local SGD:

$$x_i^{k+\frac{1}{2}} = x_i^k - \gamma(\nabla f_i(z_i^k; \xi_i^k) + N_i^k). \tag{84}$$

7:     Follows the $7^{th}$-$11^{th}$ steps of Algorithm 1.
8: **end for**

---

---

**Algorithm 3** Differentially Private Decentralized Learning with Fixed Gradient Clipping Bound (**ClipD$^2$P**)

---

1: **Initialization:** $x_i^0 = z_i^0 \in \mathbb{R}^d$, $w_i^0 = 1$, learning rate $\gamma > 0$, total number of iterations $K$, privacy budget $(\epsilon, \delta)$ and fixed clipping bound $C$.
2: **for** $k = 0, 1, 2, ..., K-1$, at node $i$, **do**
3:     Randomly samples a local training data $\xi_i^k$ with the sampling probability $\frac{1}{J}$;
4:     Computes stochastic gradient at $z_i^k$: $\nabla f_i(z_i^k; \xi_i^k)$;
5:     Clips stochastic gradient by:

$$g_i^k = \frac{\nabla f_i(z_i^k; \xi_i^k)}{\max\left\{ 1, \frac{\left\| \nabla f_i(z_i^k; \xi_i^k) \right\|}{C} \right\}}; \tag{85}$$

6:     Draws randomized noise $N_i^k$ from the Gaussian distribution:

$$N_i^k \sim \mathcal{N}\left(0, \sigma^2 C^2 \mathbb{I}_d\right), \tag{86}$$

where $\sigma$ is defined in Proposition 2;
7:     Differentially private local SGD:

$$x_i^{k+\frac{1}{2}} = x_i^k - \gamma(g_i^k + N_i^k); \tag{87}$$

8:     Follows the $7^{th}$-$11^{th}$ steps of Algorithm 1.
9: **end for**

---

## C  MISSING DEFINITION OF TIME-VARYING DIRECTED EXPONENTIAL GRAPH

We supplement the definition of time-varying directed exponential graph (Assran et al., 2019) we missed in the main text. Specifically, $n$ nodes are ordered sequentially with their rank $0,1...,n-1$, and each node has out-neighbours that are $2^0, 2^1, ..., 2^{\lfloor \log_2(n-1) \rfloor}$ hops away. Each node cycles through these out-neighbours, and only transmits messages to one of its out-neighbours at each iteration. For example, each node sends message to its $2^0$-hop out-neighbour at iteration $k$, and to its $2^1$-hop out-neighbour at iteration $k+1$, and so on. The above procedure will be repeated within the list of out-neighbours. Note that each node only sends and receives a single message at each iteration.

## D  ADDITIONAL EXPERIMENTS RESULTS

In this section, we present additional experimental results.

**Deep CNN ResNet-18 training.**    For the task of training deep CNN model ResNet-18 on Cifar-10 dataset, we run AdaD$^2$P, ConstD$^2$P and ClipD$^2$P for 3500 iterations and compare their performance under the same noise scale $\sigma$ selected from the set $\{0.001, 0.03\}$. It can be observed form Figures 5 and 6 that AdaD$^2$P always outperforms other algorithms ConstD$^2$P and ClipD$^2$P which both employ fixed-level DP noise under the same level of privacy protection, in terms of the convergence of gradient norm, training loss and model accuracy. These experiments further confirm the superiority of our proposed AdaD$^2$P employing adaptive noise level, compared to its counterparts employing fixed-level noise.

**Shallow 2-layer neural network training.**    For the task of training shallow 2-layer neural network on Mnist dataset, we run AdaD$^2$P, ConstD$^2$P and ClipD$^2$P for 2200 iterations and compare their convergence performance under the same noise scale $\sigma$ selected from the set $\{0.03, 0.04, 0.06\}$. The experiments presented in Figures 7, 8 and 9 demonstrate that: under the same level of privacy protection (same $\sigma$), AdaD$^2$P achieves superior model accuracy compared to its counterparts that employ fixed-level noise, which verifies the effectiveness of our adaptive noise mechanism.

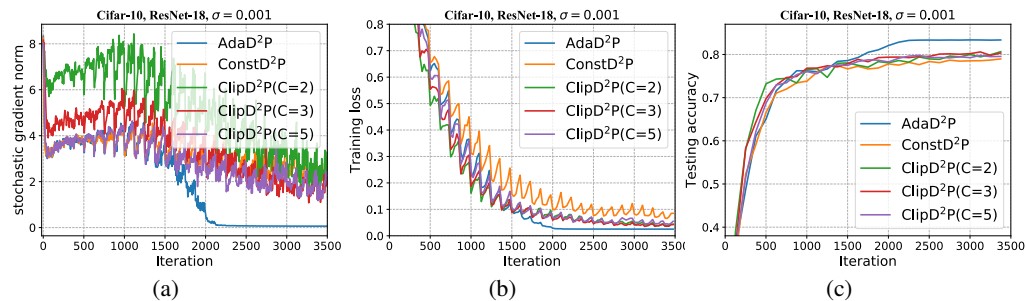

Figure 5: Performance comparison of training ResNet-18 for AdaD$^2$P with ConstD$^2$P and ClipD$^2$P under the same noise scale $\sigma = 0.001$.

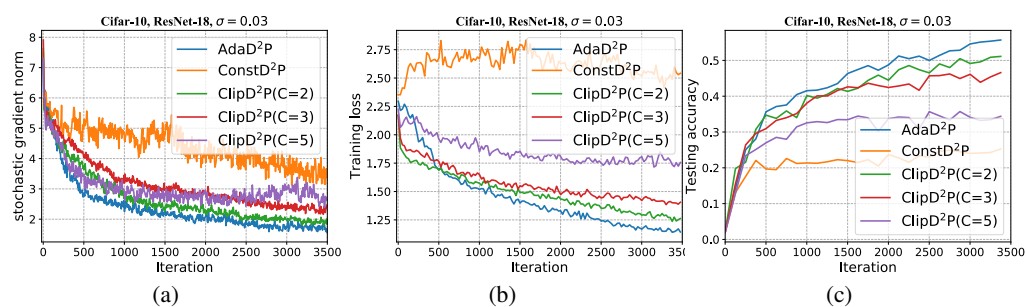

Figure 6: Performance comparison of training ResNet-18 for AdaD$^2$P with ConstD$^2$P and ClipD$^2$P under the same noise scale $\sigma = 0.03$.

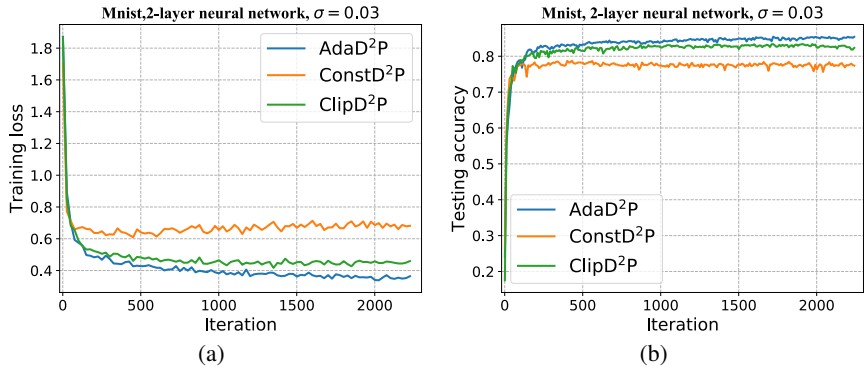

Figure 7: Performance comparison of training 2-layer neural network for AdaD$^2$P with ConstD$^2$P and ClipD$^2$P under the same noise scale $\sigma = 0.03$.

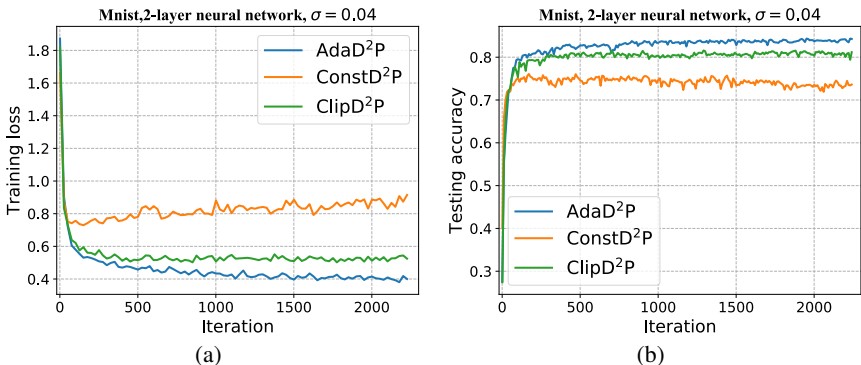

Figure 8: Performance comparison of training 2-layer neural network for AdaD$^2$P with ConstD$^2$P and ClipD$^2$P under the same noise scale $\sigma = 0.04$.

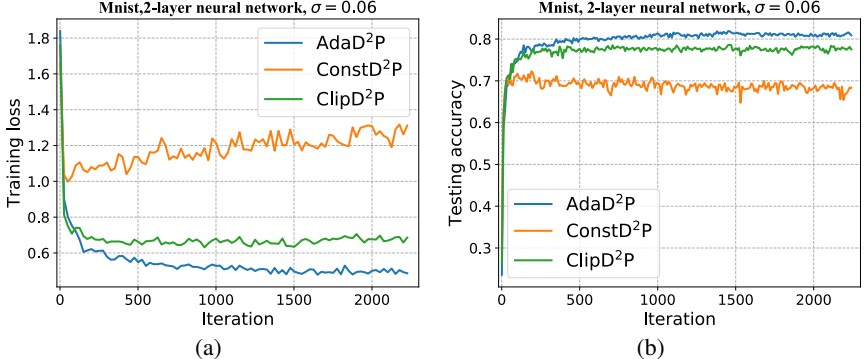

Figure 9: Performance comparison of training 2-layer neural network for AdaD$^2$P with ConstD$^2$P and ClipD$^2$P under the same noise scale $\sigma = 0.06$.