# OpenReview forum: "Sensitivity-Aware Differentially Private Decentralized Learning with Adaptive Noise"
_ICLR.cc/2024/Conference — ICLR 2024 Conference Withdrawn Submission_

### Official Review · Reviewer_edhv · 2023-10-27

**Soundness:** 1 poor
**Presentation:** 3 good
**Contribution:** 1 poor
**Rating:** 3
**Confidence:** 4

**Summary:**

This paper studies the differentially-private decentralized SGD and proposes using adaptive noise proportional to the local sensitivity, which saves a mount of noise compared to the noise bound based on the worst-case sensitivity. But such adaptive noise cannot produce worst-case DP guarantee, and thus the comparisons with prior works using clipped SGD with noise proportional to the clipping bound are not fair.

**Strengths:**

The plots are clear and the statements are clearly made.

**Weaknesses:**

1. My major concern is the adaptive noise proposed in this paper does not produce worst-case DP guarantee: local sensitivity is not equivalent to the global sensitivity and that is why existing works need to do gradient clipping to ensure this bound. Though such clipping is artificial, it is the only known way to give the sensitivity bound of general DP-(decentralized) SGD. Therefore, all the experiments with comparisons to prior works are not fair. I can only take the empirical results as a technical report to study the distribution of practical gradient/local update norm.

2. The theoretical results on both the convergence rate and the utility-privacy tradeoff seem weak, which do not provide too much new insights (largely the analysis of noisy decentralized SGD) and do not improve the prior results.

3. In the experiments, only the relationship between loss/test accuracy and noise scale is plotted while the corresponding (epsilon, delta) guarantee is not mentioned. Even under such a weak privacy guarantee (not DP), the comparison with state-of-the-art deep learning results on CIFAR10 with DP-SGD is not provided.

**Questions:**

The authors need to first fix the fundamental technical problem before the paper can be considered for publication.

---

### Official Review · Reviewer_8T7D · 2023-10-29

**Soundness:** 2 fair
**Presentation:** 2 fair
**Contribution:** 1 poor
**Rating:** 3
**Confidence:** 4

**Summary:**

This paper aims to use adaptive sensitivity in privately distributed learning to improve utility performance, which is motivated by the fact that in practice the gradient norm will vanish.

**Strengths:**

The idea of adaptively adjusting the noise according to the actual gradient norm is interesting and deserves its exploration in non-convex distributed learning.

**Weaknesses:**

1. The privacy guarantee seems not to hold.
2. The utility analysis seems to be standard.

**Questions:**

I have some concerns about the privacy guarantee in Proposition 2 (If I did not miss something obvious here)

1. Why it is DP? The noise needs to scale with the maximum change (i.e., sensitivity) to provide DP (which is the worst-case guarantee).
2. However, the current scheme (Eq. 3) only adds noise according to the specific/particular instance of gradient norm.
3. If I understand it correctly, it cannot guarantee DP (simply consider K = 1, J = 1)

---

### Official Review · Reviewer_ktTZ · 2023-10-30

**Soundness:** 2 fair
**Presentation:** 3 good
**Contribution:** 2 fair
**Rating:** 3
**Confidence:** 3

**Summary:**

This paper studies the problem of decentralised learning with DP guarantees. It proposes a algorithm that adds Gaussian noise with the noise level adaptive to the gradient norms. In particular, the authors claim that their theoretical results can remove the bounded gradient assumption and the clipping steps that are commonly used in the literature.

**Strengths:**

The problem under consideration is of great importance. The writing is mostly clear and sufficient theoretical and empirical evidences are provided. The authors also make a good effort in discussing related work.

**Weaknesses:**

There are two main components in the proposed algorithm - a non-private part that builds on the Stochastic Gradient Push algorithm proposed in Assran et al. (2019) and the part that involves privacy analysis of the Gaussian noises. The first part does not seem novel and the analysis seems to mostly follow from the previous work. The main contribution comes from the privacy analysis, about which I have a key question.

**Questions:**

Among other questions, the one that concerns me the most is that, without any clipping, how can you show that the $l_2$ sensitivity of the gradient is bounded? To be specific, can you explain why

$
||\nabla f_i(z_i^k;\xi_i^k) - \nabla f_i(z_i^k;(\xi_i^k)') ||_2 \lesssim ||\nabla f_i(z_i^k;\xi_i^k)||_2,
$

for any $\xi_i^k$ and $(\xi_i^k)'$ that differ by one entry? I think this is needed for you to add Gaussian noise with noise level proportional to $||\nabla f_i(z_i^k;\xi_i^k)||_2$, while guaranteeing $(\epsilon,\delta)$-DP.

Other comments include:

1.The definition of $l_2$ sensitivity should be provided.

2.Some notions of graph, e.g. strongly connected, diameter, in-neighbour, should be defined and discussed.

3.The Descent Lemma referred to in the proof should be stated.

4.Some remarks on the convergence criterion would be appreciated as it is not on the loss function or the parameters directly.

---

### Official Review · Reviewer_GXhs · 2023-10-31

**Soundness:** 4 excellent
**Presentation:** 3 good
**Contribution:** 3 good
**Rating:** 6
**Confidence:** 3

**Summary:**

Differentially private usually requires adding noise depending on the Lipschitz constant G since G dictates how much the gradient changes with different data. Instead of adding noise depending on G, this paper proposes a mechanism that adds Gaussian noise with variance depending on the actual gradient norm. Since the norm is small at the end of the training as the model converges to the stationary points, the amount of noise that we need to add is a lot smaller than usual. The authors then show the theoretical guarantee and evaluate the mechanism's performance in practice.

**Strengths:**

- DP-decentralized learning is an important problem that has a lot of practical implications.

- The proposed mechanism achieves the SOTA convergence rate without the Lipschitz bound assumption.

- The experiment shows improvement over the mechanism that uses Lipschitz bound and clipping.

- The paper is well written overall.

**Weaknesses:**

- I think the authors should give some intuitions or some explanations on how they handle using adapted noise without gradient bounds since it's the main technical contribution of the paper. Based on the proofs, it seems like it was handled by relating the sum of the gradient with $\|z_i^k - \bar x^k\|$, and then with careful settings of the step size, we were able to remove the gradient bounds assumptions. However, from the main text, it's not too obvious how they do it.

**Questions:**

- Does the analysis that doesn't require the Lipchitz bound work for other models? For example, the single-node centralized model or the DP model in the DP-SRM paper.

- How was the clipping constant chosen?

---

### Official Review · Reviewer_pHbp · 2023-11-01

**Soundness:** 3 good
**Presentation:** 2 fair
**Contribution:** 2 fair
**Rating:** 5
**Confidence:** 3

**Summary:**

This paper proposes a novel differentially private decentralized optimization with Gaussian noise adaptive to gradient norms. This method achieves a new utility bound under more relaxed assumption than previous literature. Some experiments on ResNet-18 and CIFAR10/MNIST confirm the performance of this method.

**Strengths:**

The paper is clearly written, especially the assumptions, and novel as far as I can tell. It is convincing to see that the authors show both empirically and theoretically that the new method works.

**Weaknesses:**

1. The experimental comparison of methods seems not complete. What about methods in Table 1 and non-DP counterparts (including yours, i.e. sigma=0, which is usually a very insightful upper bound of any DP method)? Currently only 3 methods are compared, which seems strange given that the paragraph "Decentralized learning methods with privacy guarantee" discussed so many.

2. C=0 in Figure 2? In Figure 2(bcef), it empirically looks like smaller C is better. This leads to an interesting but missing comparison to infinitely small C, i.e. gradient normalization (clipping every gradient to the same value). See "Automatic Clipping: Differentially Private Deep Learning Made Easier and Stronger" for the centralized setting and "DP-NormFedAvg: Normalizing Client Updates for Privacy-Preserving Federated Learning" for the decentralized one. Would it outperform AdaD2P?

3. The models and datasets are too toy-like. I would at least expect to see CIFAR100, which is of the same size as CIFAR10 but more difficult. It would be desirable to see also ResNet 34 or 50 (more compute, but managable by your machines), and ViT-tiny or small (similar compute as ResNet 18). Is there a foreseeable challenge to experiment on language tasks?

I would raise my score if my questions are properly addressed.

**Questions:**

See Weaknesses.